# Neurogenetic profiles delineate large-scale connectivity dynamics of the human brain

Ibai Diez[1,2] & Jorge Sepulcre[1,3]

Experimental and modeling work of neural activity has described recurrent and attractor dynamic patterns in cerebral microcircuits. However, it is still poorly understood whether similar dynamic principles exist or can be generalizable to the large-scale level. Here, we applied dynamic graph theory-based analyses to evaluate the dynamic streams of whole-brain functional connectivity over time across cognitive states. Dynamic connectivity in local networks is located in attentional areas during tasks and primary sensory areas during rest states, and dynamic connectivity in distributed networks converges in the default mode network (DMN) in both task and rest states. Importantly, we find that distinctive dynamic connectivity patterns are spatially associated with Allen Human Brain Atlas genetic transcription levels of synaptic long-term potentiation and long-term depression-related genes. Our findings support the neurobiological basis of large-scale attractor-like dynamics in the heteromodal cortex within the DMN, irrespective of cognitive state.

[1] Gordon Center for Medical Imaging, Department of Radiology, Massachusetts General Hospital and Harvard Medical School, Boston 02114 MA, USA. [2] Neurotechnology Laboratory, Health Department, Tecnalia Research & Innovation, Derio 48160, Spain. [3] Athinoula A. Martinos Center for Biomedical Imaging, Department of Radiology, Massachusetts General Hospital and Harvard Medical School, Charlestown 02129 MA, USA. Correspondence and requests for materials should be addressed to J.S. (email: sepulcre@nmr.mgh.harvard.edu)

Neurons of the human brain are assembled to form dynamic systems from local to large-scale distributed spatial scales. Dynamic activity and oscillatory synchronization of neurons shape the communication among cerebral areas and are thought to generate complex self-organizing and adaptive patterns of functional connections[1]. While enormous advances in the study of spikes and focalized microcircuits have led to insights regarding the dynamic behavior of neurons, the features that support cooperative dynamic communications of the human brain at a larger scale are less well understood. For instance, the characterization of the temporal changes of functional connectivity networks remain elusive, and there are no commonly accepted notions about how self-organizing collective interactions of neurons emerge at the large-scale brain level[2,3]. Thus, investigations are needed to understand the dynamic patterns supported by human whole-brain functional connectivity.

Since Poincaré's seminal advances on recurrence of dynamic systems and phase space, there has been great interest in investigating the temporal flow of activity and connectivity in the brain by analyzing the non-linear transitions of discrete variable states[2]. Models based on experimental work have revealed several dynamic properties of neurons. Neural circuits forming directed cycles exhibit repetitive or recurrent temporal dynamics and attractor behaviors—a dynamic pattern that a system tends to evolve or settle into[4,5]. The study of recurrent dynamic patterns has also expanded from the microscopic scale to the meso- and macroscopic levels of analysis to improve our understanding of the intrinsic dynamics of large and distant regions of the brain[6]. Findings from electroencephalogram (EEG) experiments have also pointed to the existence of recurrent, reverberant or attractor patterns—such as limit cycle and fixed-point attractors[7,8]—that in turn might explain the large-scale multi-stable synchronicity of the brain. Other studies have associated the sustained EEG oscillations in epileptic seizures with transitions toward limit cycle or chaotic attractors[7,9].

Thus, the present study aimed to characterize the brain configurations that show repeated dynamic connectivity toward precise locations in the cortical mantle during multiple task performances. Building upon previous descriptions in computational modeling that utilize structural and functional neuroimaging techniques[10,11], we propose an empirical and data-driven approach to describe the underlying dynamic patterns of large-scale functional connectivity networks in two ways. First, we characterized large-scale connectivity changes in time by localizing nodes that display a high degree of functional streams (or dynamic paths on graphs) converging to specific points of the cortical space and across multiple brain states as a proxy of attractor-like behavior. We conjectured that there are network nodes that are targeted by other nodes of the brain network, in which information streams are repeatedly formed towards them. Second, we investigated whether dynamic connectivity of large-scale brain networks are founded on specific cellular and molecular mechanisms through neuroimaging–genetics expression interactions in the human cerebral cortex[12–15].

## Results

**Large-scale dynamics of human brain functional connectivity**. Our analysis of dynamic connectivity in task performances using stepwise functional connectivity (SFC) as a proxy for recurrent network behaviors (Fig. 1) yielded distinct findings for local and distributed connectivity maps (cortical maps in Fig. 2). Local dynamic connectivity during task performances converged repeatedly in areas associated with attentional processes and task monitoring such as the lateral occipital, frontoparietal or dorsal attention networks (cortical maps in Fig. 2a; line graphs in Fig. 2a;

note that line graphs represent the SFC values or connectivity paths that reach specific voxels repeatedly). In contrast, distributed dynamic connectivity during wide task performances predominantly converged in regions of the default mode network (DMN), particularly in the precuneus/posterior cingulate cortex and midline prefrontal areas (cortical maps in Fig. 2b). During the time course of each task, local dynamic connectivity accumulated a disproportionate number of local streams that reached attentional and task-related areas, while changes in distributed connections showed that global streams of connectivity consistently reached prominent areas of the DMN in all time windows for all types of tasks (line graphs in Fig. 2b; note that line graphs represent the SFC values or connectivity paths that reach specific voxels repeatedly). Interestingly, the distributed dynamic connectivity map can only be achieved after the exclusion of modular connections, while the topology of the local dynamic connectivity map is equivalent to the map obtained when all connectivity is included in the analysis (see Fig. 2a and Supplementary Fig. 3).

The results of our analyses assessing local dynamic connectivity during task performances and resting-state yielded distinctive cortical patterns (cortical maps in Fig. 3). While local task-related connectivity converged in attentional and monitoring areas, local resting-state connectivity tended to dwell in the somatomotor, visual and auditory cortices (cortical maps in Fig. 3a). Conversely, we found that distributed recurrent dynamic connectivity during both task and resting-state allocate similarly at the spatial level, involving areas of the DMN (cortical maps in Fig. 3b). Similar results were obtained for local and distributed dynamic connectivity with the two replication datasets (Pearson's correlation of 0.98 for local and 0.90 for distributed, see Supplementary Fig. 6). Moreover, we observed that the strength of global dynamic connectivity streams converging in the DMN forced those participating regions to be closely together—although without triangle motifs (see methods for details)—within the topological space in a consensus task connectivity network (network graph in Fig. 3c). Finally, as our SFC approach is able to describe dynamic trajectories of connectivity on graphs, we also analyzed specific trajectories of dynamic connectivity using the original and replication datasets. We evaluated the cortical areas with specific SFC values and tested if dynamic trajectories of paths remain inside or go outside those areas. We found that cortical areas with high local and distributed dynamic connectivity (or SFC values) tend to display dynamic paths that remain repeatedly inside those areas, while regions with low SFC values display dynamic paths that go toward cortical areas with high SFC values (Fig. 4).

**Dynamic connectivity and its cortical genetic signature**. To understand the neurobiological basis of large-scale dynamic connectivity, we compared the average maps of Fig. 3 with the cortical expression of neuro-related genes (~3700) from the Allen Human Brain Atlas. We obtained the spatial similarity between each comparison (see histograms in Fig. 5a, b). We found the gene expression levels of 211 and 195 genes were distributed along the cortical mantle similarly as the local and distributed connectivity maps, respectively (>1.65 SD). Using false discovery rate (FDR)-corrected Gene Ontology (GO) overrepresentation analysis (FDR-corrected $q < 0.005$), we found that genes associated with the local dynamic connectivity map displayed an overrepresentation of action potential and ion channel-related genes, while genes associated with the global dynamic connectivity map were more functionally engaged in long-term potentiation (Fig. 5 and Supplementary Tables 2 and 3). Other overrepresented functional annotations from both lists of genes

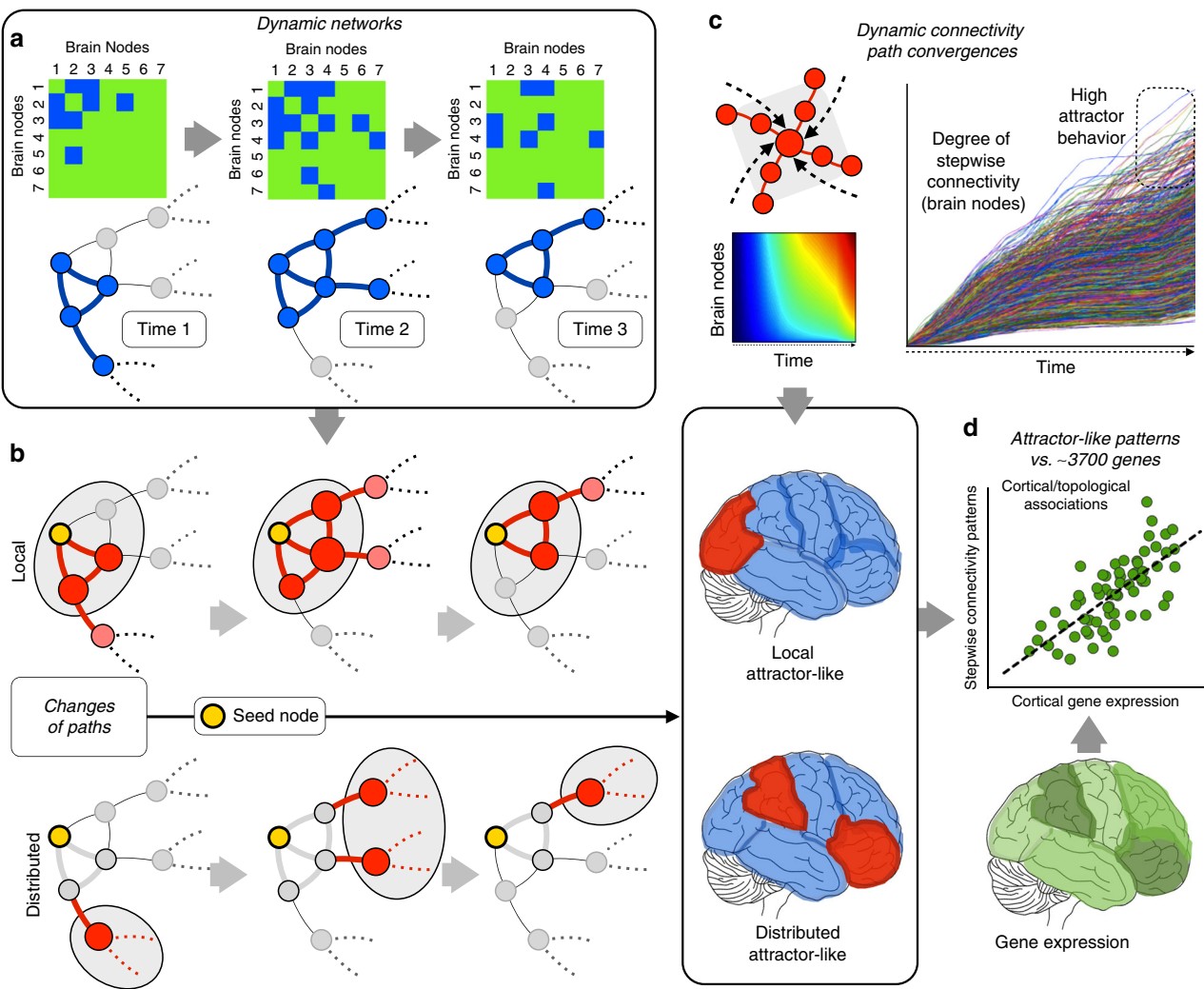

**Fig. 1 a** Dynamic connectivity was evaluated using network configurations over windows of time through association matrices and graph changes. Three binary examples of time points are displayed for illustration purposes, although the real data are computed as weighted graphs. **b** To describe the local patterns of dynamic connectivity as well as distributed network changes outside local modules, we separately investigate the direct and indirect neighbors of all given investigated nodes in the entire cortex. As seen in the diagrams, this strategy enables the comparison of modular (or local) connectivity and distant connectivity. **c** SFC is used to investigate the functional streams that converge at specific points of the cortex (network phase space). Top left: diagram of converging streams in a given node. Top right and top center: calculation of the recurrence of streams targeting voxels over time at the whole-brain level (lines graph and density graph). Bottom: Due to the segregation between local and distributed networks, we can independently investigate the local and distributed recurrence connectivity. The diagrams represent the cerebral cortex and red areas with high accumulative degree of recurrent streams hitting them over time. **d** Local and distributed recurrence connectivity maps are compared with neuro-related genes of the human transcriptome (theoretical example shown as the green cortical map) via spatial similarity using a linear regression approach (scatterplot)

were associated with neuron and axon development and synaptic transmission (bar graphs in Fig. 5a, b). Interestingly, when we investigated the specific long-term potentiation (LTP) genes (as well as the counterpart long-term depression (LTD) genes) associated with dynamic connectivity maps, we found that the local map involved 4 LTP (*CA7*, *GRIN2A*, *STX1B*, *SYT12*) and 3 LTD (*KCNB1*, *PNKD*, *PICK1*) genes, while the distributed map included 9 LTP (*CHRNB2*, *CRH*, *EGFR*, *NTRK2*, *LGI1*, *PRKCE*, *PRNP*, *RIMS1* and *STX1B*) and 1 LTD gene (*SLC30A1*) (>1.65 SD, see histogram in Fig. 5 and the Venn diagram in Fig. 6a). Using a stringent cutoff (1.96 SD), we found that the local map involved 1 LTP (*SYT12*) and 1 LTD (*KCNB1*) gene, while the distributed map included 6 LTP (*CRH*, *LGI1*, *PRKCE*, *PRNP*, *RIMS1* and *STX1B*) and 1 LTD (*SLC30A1*) genes (Venn diagram in Fig. 6a). Finally, we found that the spatial association patterns between cortical maps of dynamic changes and three of these genes displayed multiple comparison corrected significant

differences between the local and distributed patterns (Fig. 6b; familywise error rate (FWE) <0.05). Namely, we found that the spatial relationships between the local map and *KCNB1* (LTD) and *SYT12* (LTP) genes are significantly different than the same spatial relationships with the distributed map ($p = 0.0011$ and $p = 0.0002$, respectively). Moreover, the spatial relationship between the distributed map and the *PRNP* gene (LTP) is significantly different than the same spatial relationship with the local map ($p = 0.0031$).

## Discussion

In this study, we studied empirical functional connectivity data over time during various cognitive states to reveal the zones of the human brain in which convergence of recurrent connectivity occurs. Our aim was to detect and map the specific areas engaged in attractorness-like behavior at the local and distributed levels.

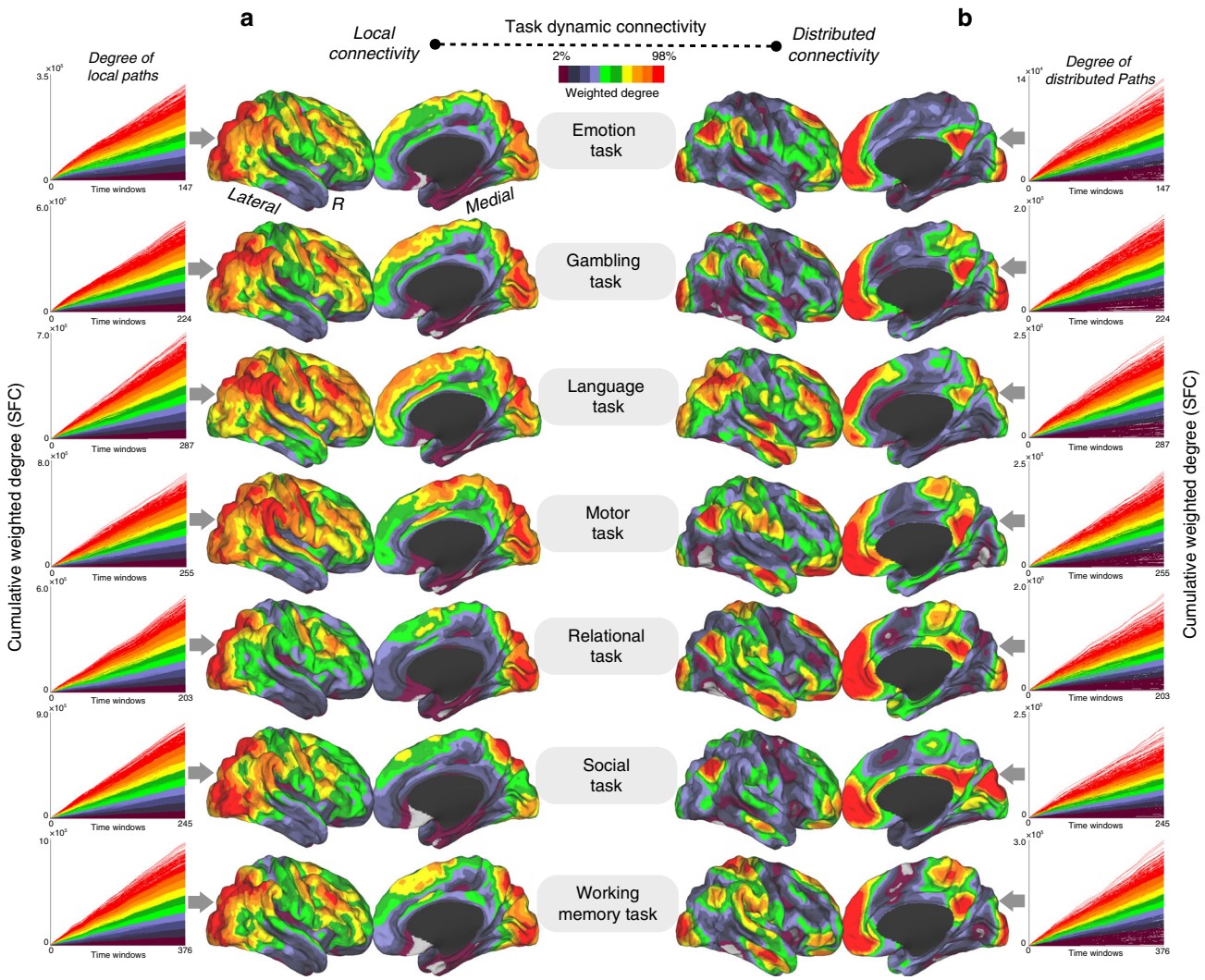

**Fig. 2 a**, **b** For each cognitive task, the accumulative SFC score capturing the functional connectivity streams that reach voxels over time is shown, as is the average cortical maps of the dynamic connectivity patterns and in the form of line graphs (**a** local dynamic connectivity and **b** distributed dynamic connectivity). Color scale in line graphs and cortical maps represent 2–98% of the normalized weighted degree of SFC

As previously suggested, we assumed that the human brain has numerous recurrent dynamic sources and networks that may produce multiple attractors in a multi-stability scenario[6,10]. Neural activity does not occur in isolation but is synchronized with other neuronal signals. This organization tends to repeat over time, and some coupled regions tend to be orchestrated more frequently and recurrently than others. Connectivity between brain areas via phase synchronization forms functional networks, and dynamic and transient patterns arise from the cooperation and competitiveness among them. Moreover, recurrent activity across neural networks is thought to yield self-organized and multi-scaled dynamics patterns in the human brain[2,11,16,17]. At the spatial level, it has been demonstrated that flows of activity spread from specific areas toward certain local or distant locations of the cortex. In general, this property of brain activity streaming repeatedly toward precise locations can be seen or conceptualized as an attractor or attractor-like behavior. In the past, neural network modeling has reproduced feasible scenarios of recurrent or attractor-like dynamic patterns at the synaptic and neuronal levels. Since the introduction of the concept put forth by Lorente de No and Hebb of reverberation as neural activity that reiterates in a network, researchers have studied the implications of recurrent neuronal activity in cell assembly formation and

cellular memory processes in brain circuits, which are key components of cortical networks[18]. However, it is still poorly understood how these types of dynamics transfer or generalize to larger spatial scales and whether self-organized patterns, such as reverberancy/recurrence or attractorness, arise from the functional connections of the human cortex. The existence of self-organized and attractor dynamic patterns in the human brain networks has been postulated to be critical to our understanding of how cognitive processes, behavior, action–perception cycles or mind–brain–body integration form in humans[16,19,20]. Compared to previous studies, our study employed a data-driven approach that goes directly from empirical functional connectivity magnetic resonance imaging (fcMRI) data to the investigation of the biological basis supporting large-scale dynamic patterns of the human brain. By doing so, we show that the DMN displays dynamic connectivity and genetic features that favors it as the main attractor network of the human brain.

As recently stated by Friston[21,22], "to properly understand neural processing, one has to move beyond classical information theoretic approaches and look at the itinerant dynamics that underpin self-organization in nonequilibrium systems, like the brain, that maintain a steady state or homeostasis". Previously, many studies have found that signal equation modeling and post

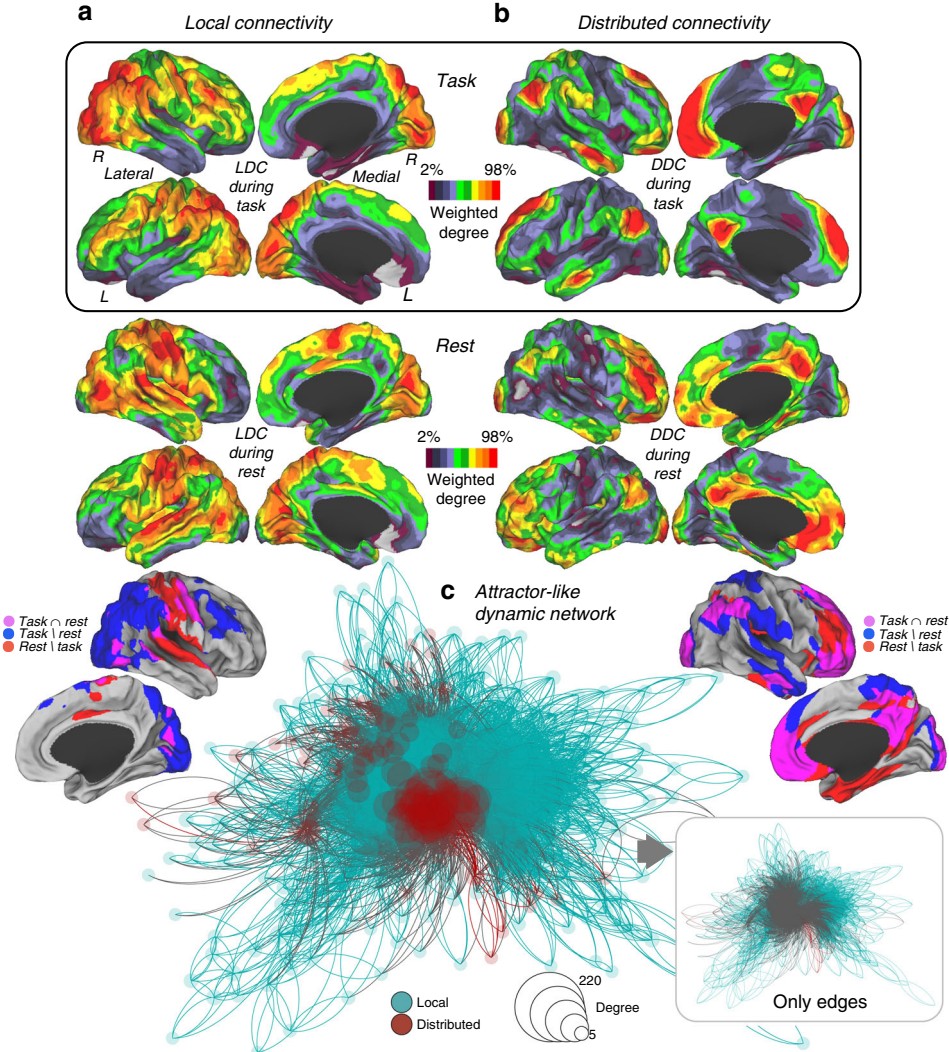

**Fig. 3 a**, **b** Average maps of all task domains and overlap map with resting-state dynamic connectivity patterns. **c** Local and distributed connectivity in the network topological space is displayed. The size and color of the nodes are based on the degree of local and distributed connectivity. The degree centrality (degree of connectivity) shows higher values as larger nodes. The size and color of links are based on the subtraction between local and distributed SFC scores. LDC local dynamic connectivity, DDC distributed dynamic connectivity. Local is in blue color and distributed red color

hoc validation with EEG or functional MRI data supports the existence of attractor behavior of the large-scale human brain[11,21]. An attractor system describes dynamic trajectories over a specific phase space that is considered structurally stable when perturbations produce slight modifications in its shape. In this context, we can speculate that if multiple stable attractors co-exist in the human brain functional connectome with their specific attraction basins, we should be able to describe repetitive patterns of dynamic synchrony and functional connectivity streams over time in specific cortical locations. In contrast, if brain attractors are highly unstable and with many bifurcation points (criticality), then our ability to detect consistent repetitive behaviors in specific points of the space should diminish. To investigate this hypothesis, we studied many different brain states (or natural perturbations associated with different task performances) in order to capture multiple attractors. In other words, we investigated the connectivity patterns that are stable and recurrent across many cognitive states. As a result of our analyses, we found that local networks in modal, as previously suggested[23], and attentional cortices—dorsal and ventral attentional systems

as well as in the frontoparietal network—exhibit attractor properties in different cognitive states, while distributed attractors in the heteromodal cortex are present and stable regardless of cognitive states.

Although much is recognized about the connectional properties of the functional connectome, the neurobiological and genetic foundation of its dynamic mechanisms are less well known. In this work, we showed that recurrence connectivity in local networks relate to a mix of synaptic potentiation and neuronal excitability genes, particularly *KCNB1*, a voltage-gated potassium channel related to LTD and the regulation of neurotransmitter release[24], as well as *SYT12*, a member of the synaptotagmin gene family that mediates LTP, synaptic-vesicle exocytosis and calcium-dependent regulation of synaptic trafficking[24]. Thus, our cortical pattern of attractorness in local networks match the cortical gene expression of several genes related to synaptic potentiation in both LTD and LTP profiles. These findings are in agreement with recent studies reporting associations between cortical synchronous activity and genetic expression of ion channel functionality[15]. Additionally, it has been previously

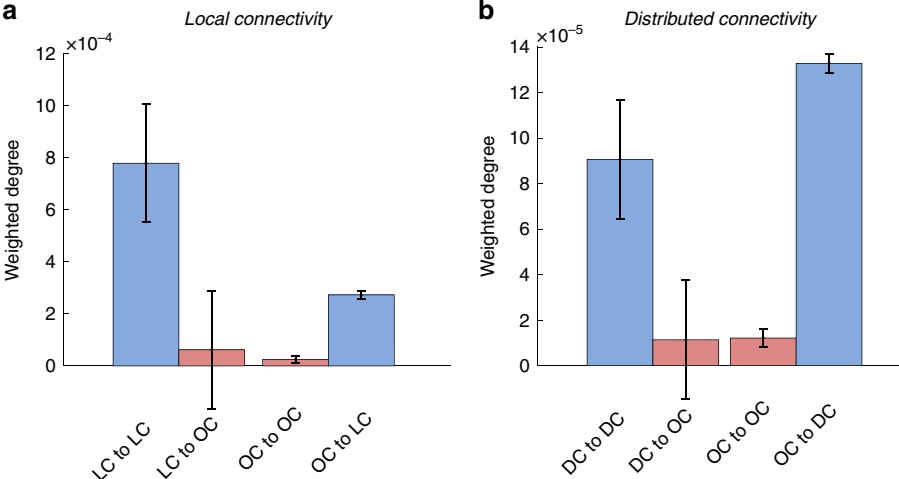

**Fig. 4** Description of dynamic trajectories of paths from cortical areas discovered as local and distributed connectivity cores during task performances. Masks from cortical areas with predominant (Fig. 3c; blue/magenta in cortical masks) and non-predominant (Fig. 3c; none blue/magenta colors in cortical masks) local (**a**) and distributed (**b**) connectivity were used to obtain the amount of dynamic trajectories that remain inside or leave outside these connectivity cores in an independent sample of individuals. "LC to LC" and "DC to DC" refers to connectivity trajectories that start and end within the predominant/core areas of local and distributed connectivity. "LC to OS" and "DC to OS" refers to connectivity trajectories that start in the predominant/core areas but end outside (OS) them. "OS to LC" and "OS to DC" refers to connectivity trajectories that start outside the predominant/core areas but end inside them. Error bars represent standard error. The y-axes represent the weighted degree of local or distributed paths (normalized by the size of cortical masks)

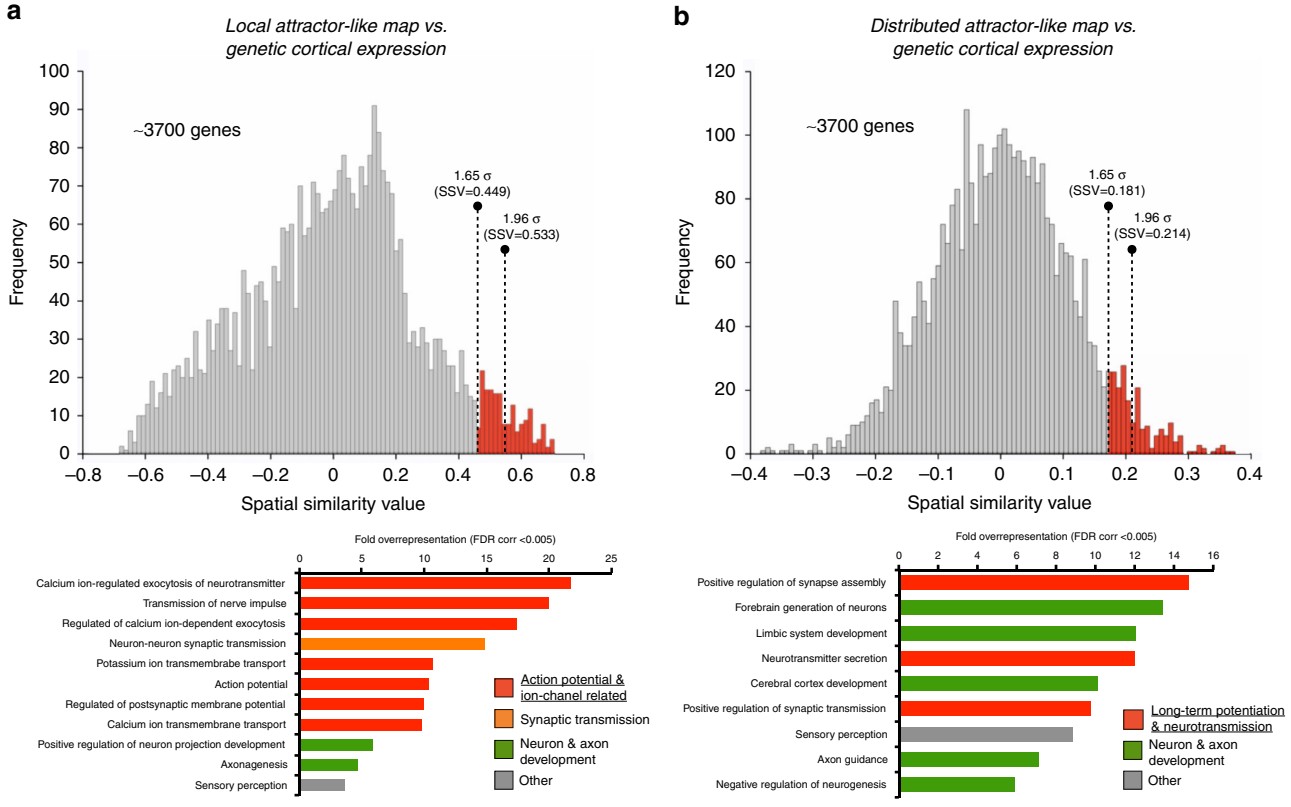

**Fig. 5** Histograms represent the distribution of spatial similarity scores between dynamic connectivity maps (local (**a**) and distributed (**b**)) and the cortical expression of neuro-related genes (~3700) from the Allen Human Brain Atlas. Two levels of statistical threshold are shown using red bars with dotted lines (>1.65 SD and >1.96 SD). Bar graphs show the FDR-corrected GO overrepresentation analyses of genes with genetic expression spatially related to dynamic connectivity maps (local (**a**) and distributed (**b**))

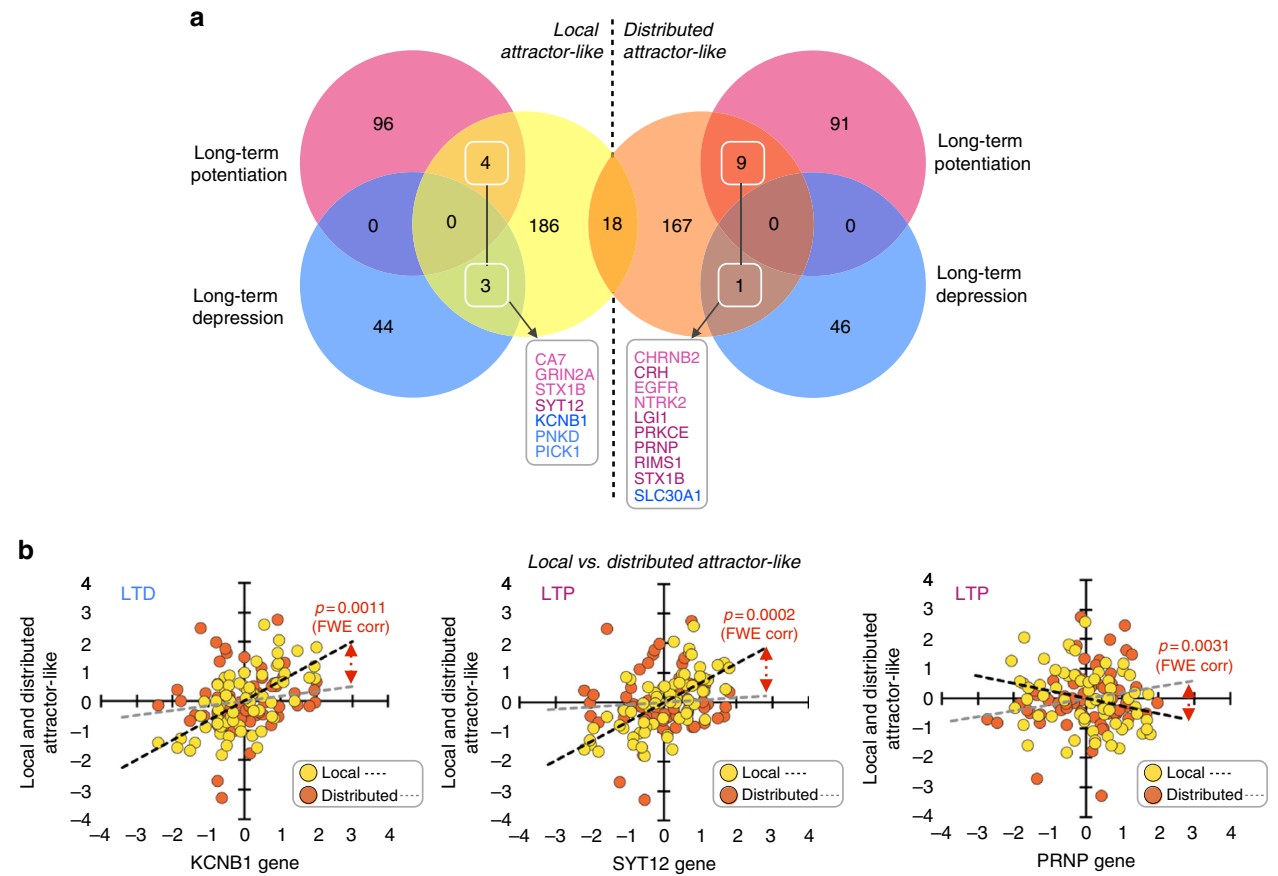

**Fig. 6** Venn diagram showing genes involved in LTP and LTD processing within the genetic background found to be associated with dynamic connectivity maps (boxes in diagram; left shows local, right shows distributed). Scatterplots show the significant differences between the local connectivity–genetic and global connectivity–genetic associations in KCNB1 (LTP), SYT12 (LTD) and PRNP (LTP) genes (FWE correction <0.05)

demonstrated that resting-state connectivity profiles of resting-state MRI data relate to the gene expression levels of ion channel genes such as *DRD2*, *GLRA2*, *HTR2C*, *KCNS1*, *KCTD4* and *SCN1B*[14]. As experimentally postulated, the dynamics of neuronal recurrence or attractors arise from the synaptic potentiation[25]. In our study, we included a genetic analysis to better understand LTP- and LTD-related gene functions and found that both are critical to explain the dynamic profiles of local networks in which we can postulate that a versatile level of flexible construction and destruction of connections (depending on whether the individual is currently engaged in primary modal, attentional or task-related processes) is needed.

Importantly, we found that the distributed recurrence of dynamic connections in the DMN relies on a distinctive genetic profile that mostly depends on LTP genes (such as *PRNP*), which encodes a membrane glycosylphosphatidylinositol-anchored glycoprotein associated with Creutzfeldt–Jakob disease, fatal familial insomnia, Gerstmann–Strausslier disease, Huntington disease-like 1 and kuru, all neurological diseases with signs of sustained aberrant neuronal activity[24]. Moreover, following Hebb's speculation that transient memories are maintained by reverberation until stable synapses are acquired[18], it is possible to support the idea that networks with dynamic recurrence and LTP genetic expression are the substrate for consolidation of cellular memories in the human cortex. Thus, a predominant LTP profile may be relevant to induce consistent connectivity recurrence at the distributed spatial scale and to support a potential attractorness behavior regardless of cognitive state. Based on our dynamic connectivity findings in which the DMN remains constantly recurrent across all brain states at the distributed level, we

postulate that the DMN has a main role in stabilizing the entire large-scale system. This interpretation is consistent with the especulative role of the DMN as a "global workspace" and large-scale brain integrator[19,26–30]. In the past, modeling approaches have supported DMN functionality as a global integrator or facilitator to communicate different network states, mostly via the cortical hubs or rich club. For instance, previous findings show that the emergence of resting-state patterns reflects the exploration of network configurations around a large stable network core[10,31–34]. Our results support that distributed dynamic connectivity in the DMN may favor those easily reachable and stable global brain configurations.

In this study, we provide evidence of the dynamic organization of recurrent connectivity and its neurogenetic transcriptomic signatures in the human brain. Our findings show a multi-stability scenario of the human whole-brain functional connectivity in which the temporal patterns of connectivity tend to converge into specific points of the connectome space at the local and distributed levels. Importantly, they also support the notion that the DMN plays an important role as a global attractor-like network across cognitive states, in which network configurations in heteromodal cortices confer dynamic properties in search of stability when large distances and distributed connectivity are engaged. Therefore, if only distributed connectivity is under consideration, the human brain may tend to display a uni- rather than a multi-stable dynamic system. In this study, we did not investigate whether the human brain shows metastability states, which is defined as a winner-less dynamic between networks forms—"a subtle blend of integration and segregation"[17]—and future research is needed in this sense. Moreover, the large-scale

nature of our study precludes us from making inferences about how the macroscale recurrence integrates with the micro-neuronal level, although our neuroimaging–genetic findings shorten the gap between the two.

## Methods

**Overview.** In this study, we used a graph theory approach called SFC[12,35] to analyze dynamic functional connectivity data in which local and distributed connectivity has been previously segregated (Fig. 1a, b). Given that the dynamic properties of an attractor system are better characterized under multiple perturbation conditions, we studied the brain under multiple task conditions to explore many natural perturbation conditions of the functional connectome. Using SFC, we studied whether a dynamic system shows converging and repetitive behavior toward specific network coordinates during different states of the network phase space. We first obtained discrete network configurations over time (Fig. 1a). Then, for each node in the brain, we separated the connectivity that belongs to its direct neighbors (the local dynamic connectivity condition) and beyond those neighbors (the distributed dynamic connectivity condition) (Fig. 1b), as well as the condition without connectivity segregation (see the section "Stepwise functional connectivity analysis" for details). As neighboring nodes engage in high connectivity strength, local connectivity overshadows distributed dynamic patterns emerging from distantly connected areas. Therefore, it is relevant to separate these two types of information in order to study modifications within local neighborhoods as well as long-range network dynamics between cortical regions. This strategy has proven to be effective to study large-scale connectivity organizational principles[12,36]. After this step, we calculated the functional streams and degree of paths that repeatedly reach each node of the brain (Fig. 1c). Finally, in order to gain insights into their biological meaning, we investigated the association between local and distributed recurrent dynamic connectivity patterns with the cortical gene expression levels from the Allen Human Brain Atlas (Fig. 1d)[37] (http://human.brain-map.org; neuroimaging implementation by French and Paus[38]).

**Participants.** All neuroimaging data were provided by the Human Connectome Project, WU-Minn Consortium (Principal Investigators: David Van Essen and Kamil Ugurbil; 1U54MH091657) funded by the 16 National Institutes of Health (NIH) Institutes and Centers that support the NIH Blueprint for Neuroscience Research; and by the McDonnell Center for Systems Neuroscience at Washington University. A total of $N = 30$ randomly selected healthy controls were included in this study; 14 males and 16 females between 22 and 35 years old. High-resolution T1 anatomical images, and functional magnetic resonance images (fMRI), later converted into fcMRI, from the Human Connectome Project were used in this study. Apart from the main sample, we included two additional independent samples of 30 individuals each from the Human Connectome Project (both with 17 females and 13 males between 22 and 36 years old) for replication purposes. For more information on the acquisition parameters, see Supplementary Methods and the Human Connectome Project documentation (http://www.humanconnectome.org/).

**Experimental design.** All task and rest conditions were designed and performed as part of the Human Connectome Project (http://www.humanconnectome.org/). For specific details about the experimental designs, see Supplementary Methods. Subjects were asked to complete 7 different cognitive tasks and 1 rest task inside the MR (Supplementary Table 1). Emotion processing (valenced facial masking task): Participants were presented with two faces (expressing anger or fear) or two shapes at the bottom and one at the top of the presentation and were asked to choose which of the two images at the bottom match the image at the top. Gambling (risk/reward): Participants guessed the number on a mystery card in order to win or lose money. To do so, they were asked to indicate if they thought the mystery card number was greater or less than 5 by pressing designated buttons. They were then given feedback indicating the number on the card and size of the reward or loss. Language (story and symbol representation): Participants interleaved blocks of a story task (brief auditory stories and a question about the topic) and a math task (requires subjects to complete addition and subtraction problems). Motor: Participants were presented with visual cues that asked them to either tap their left or right fingers, squeeze their left or right toes or move their tongue to map motor areas. Relational processing: Participants were presented with two pairs of objects and were asked to decide what dimension (either shape or texture) differed across the first pair of objects before deciding whether the second pair of objects also differs along that same dimension. Social cognition: Participants were presented with objects (squares, circles, and triangles) that interacted in some way or moved randomly on the screen. Participants then stated whether the objects had a mental interaction or if they were not sure if this interaction existed. Working memory: Participants were presented with pictures of places, tools, faces and body parts followed by a 2-back working memory task and 0-back working memory task. Rest condition: Participants were instructed to simply rest with their eyes open—with relaxed fixation on a projected bright cross-hair on a dark background and presented in a darkened room—without falling asleep or thinking about anything in particular.

**Stepwise functional connectivity analysis.** Conventional fcMRI approaches derive connectivity information from the entirety of the blood oxygen level dependent (BOLD) time series and result in a time-averaged brain network graph. However, as brain network dynamic changes occur at a higher temporal scale, other strategies have been used to take full advantage of the non-stationarities that reside in temporal fcMRI information contained in the fcMRI data[39]. For instance, the sliding window approach extracts the dynamic interactions between brain areas by using a time moving window along the BOLD time series. As demonstrated by a recent study using simultaneous calcium and hemodynamic signals, short time windows represent transient neuronal co-activation that allows the capture of more information about different brain states compared to static connectivity[40]. In this study, we examined different window lengths and customized high-pass filtering to investigate dynamic connectivity patterns. At the conceptual level, shorter window lengths provide higher temporal resolution but the estimated correlation coefficients are noisy and prone to error. Longer window lengths, on the contrary, might yield more precise estimates, but lack temporal fidelity and tend toward the time-averaged solution[41,42]. Supplementary Table 2 shows all window sizes used in the analyses, from 30 to 60[41]. In the main sections of this study, we present findings in which a window size of 30 s each (TR = 0.72; 42 time points) was used to split the fcMRI data, with 1 lagged time point between them[17]. Our overlapping criterion was designed to obtain smooth transitions between network states. Before splitting the time series into different windows of 30 s, a high- and low-pass filter with a cutoff frequency of 0.01 Hz and 0.08 Hz was applied to remove spurious fluctuations[42,43] (alternative window sizes and high-pass filters in Supplementary Fig. 4; static condition, no sliding window approach Supplementary Fig. 5). The Pearson's correlation of the time series of all the voxels in each time window was computed, which generated a functional connectivity matrix for each time window. We used a whole-brain mask—containing cortical gray matter, subcortical structures and cerebellum—of 5138 voxels to extract the BOLD time series and applied the sliding window approach. This step generated a 5138 × 5138 association matrix per sliding window (sliding windows varied between task ranging from 147 to 376). All the connections with negative correlation values or correlation values with a $p$ value less than 0.05 were removed from the functional matrices to eliminate the network links with poor interpretability and low temporal correlation, which are likely to be attributable to noise[44]. Finally, we applied a variance-stabilizing transformation (Fisher's transformation) to all correlation coefficients of association connectivity matrices as a final step before our graph theory-based analysis ($c$ in equation condition 1 to 3)[45].

As recently described, network algorithms offer new opportunities to understand the brain network structure by exploring the transversal connectivity patterns across multiple relay stations[6,46–49]. For instance, they have increased our knowledge of the structural and functional hierarchies and hubs organization (cortical core of hubs and rich club) that integrate large-scale networks in the human brain[13,35,50–52]. In this study, in order to detect recurrent patterns of connectivity streams, we used a graph theory method sensitive to network path connectivity changes over time. Given that dynamic connectivity changes are expected to occur at local and distributed levels, we computed the dynamic version of the SFC method in each voxel of the brain under three separate conditions including (1) local and distributed connections without network topological segregation (Eq. 1; also called total connectivity (TC) in the text); (2) only direct or immediate neighborhood connections via triangle motifs (Eq. 2; also called local or modular connectivity (LC) in the text; Fig. 1b); and (3) connections outside the local neighborhood or triangle motifs (Eq. 3; also called distributed connectivity (DC) in the text; Fig. 1b). Therefore, it is important to remark that our local and distributed terms relates to network-based topology and not to Euclidean distances within the human brain.

(Eq. 1: TC)

TC for node $i$ is computed as:

$$S_1(i,j) = \frac{c(i,j) - \min(c)}{\max(c) - \min(c)}$$

$$S_s(i,j) = \sum_{k=1}^{n} \frac{S_{s-1}(i,k) - \min(S_{s-1})}{\max(S_{s-1}) - \min(S_{s-1})} \frac{c(k,j) - \min(c)}{\max(c) - \min(c)} [i \neq j, s > 1],$$

$$TC(i) = \sum_{s=1}^{7} \sum_{j=1}^{n} \frac{S_s(i,j) - \min(S_s)}{\max(S_s) - \min(S_s)}$$

(1)

where $c$ is the association connectivity matrix, $n$ is the number of nodes (voxels) in association connectivity matrix and $S_s$ represents the normalized stepwise connectivity matrix for step $s$. TC is the weighted degree or sum of all stepwise connections per node.

(Eq. 2: LC)

LC for node $i$ is computed as:

$$SL_1(i,j) = \frac{c(i,j) - \min(c)}{\max(c) - \min(c)} [S_2(i,j) \neq 0]$$

$$SL_s(i,j) = \sum_{k=1}^{n} \frac{SL_{s-1}(i,k) - \min(SL_{s-1})}{\max(SL_{s-1}) - \min(SL_{s-1})} \frac{c(k,j) - \min(c)}{\max(c) - \min(c)} [i \neq j, s > 1],$$

$$LC(i) = \sum_{s=1}^{7} \sum_{j=1}^{n} \frac{SL_s(i,j) - \min(SL_s)}{\max(SL_s) - \min(SL_s)}$$

(2)

where $c$ is the association connectivity matrix, $n$ is the number of nodes (voxels) in association connectivity matrix and $SL_s$ represents the normalized local stepwise

connectivity matrix for step $s$. LC is the weighted degree or sum of all stepwise connections per node.

(Eq. 3: DC)

DC for node $i$ is computed as:

$$SD_1(i, j) = S_4(i, j)[S_1(i, j) = S_2(i, j) = 0]$$
$$SD_s(i, j) = \sum_{k=1}^{n} \frac{SD_{s-1}(i, k) - \min(SD_{s-1})}{\max(SD_{s-1}) - \min(SD_{s-1})} \frac{c(k, j) - \min(c)}{\max(c) - \min(c)} [i \neq j, s > 1]$$
$$DC(i) = \sum_{s=1}^{6} \sum_{j=1}^{n} \frac{SD_s(i, j) - \min(SD_s)}{\max(SD_s) - \min(SD_s)}$$

$$(3)$$

where $c$ is the association connectivity matrix, $n$ is the number of nodes (voxels) in association connectivity matrix and $SD_s$ represents the normalized distributed stepwise connectivity matrix for step $s$. DC is the weighted degree or sum of all stepwise connectivity per node.

The network topological segregation in triangle motifs allows for the investigation of local modules as well as connectivity outside these modular components. Therefore, all SFC analyses were applied to the same data with the exception of including all (total connectivity), only local neighborhood (local dynamic connectivity) or excluding local neighborhood (distributed dynamic connectivity) connections. Of note, the original description of SFC analysis was developed in time-averaged conditions and used seeds of interest to reveal connectivity transitions across specific systems[12,35]. In the present investigation, we extended the SFC method toward the analysis of all possible voxels and time windows of the fcMRI data. Because of the occurrence of coordinated and synchronized connections in time across the entire human brain, we used whole-brain analysis to capture predominant profiles of SFC. Thus, for each functional matrix corresponding to each time window, both the local SFC and distributed SFC were computed. This process is done simultaneously for all voxels in the mask in each condition (total, local and distributed SFC) via matrix multiplication in Matlab (v8.0, The Mathworks Inc., Natick, MA). Mean SFC detects the concurrently time-dependent connectivity changes and finds nodes in which multiple-step paths (also refer here as functional connectivity streams or just functional streams) converge repeatedly across windows (Fig. 1c). Importantly, our SFC approach is a weighted strategy in which the strengths of all connections are included in the analysis (Fig. 1a shows a binary example for illustration purposes only). For each time window, we used the association matrix to calculate the connectivity steps from each node to the rest of nodes in the graph topological space until seven steps were completed. We selected seven connectivity steps as our sequence detection criteria based on previous findings on diameter and path lengths between pairs of nodes in functional connectivity graphs[35], in which more than seven steps does not improve the communicability between nodes (see Supplementary Fig. 1 for a visualization of 1 to 7 steps using a single visual seed in the occipital cortex in the Emotion Task; note that this is for illustration purposes because our approach computes the SFC simultaneously in all gray matter seeds of the human brain). Importantly, there are two strategies to calculate SFC in sliding windows. The calculation of connectivity steps of a given voxel toward the rest of the voxels can be obtained within each discrete sliding window (1 to 7 connectivity steps within the same association matrix of a given window) or across multiple sliding windows (1 to 7 connectivity steps transversally in consecutive association matrices). We tested both strategies in this study. As they yield analogous outcomes (see Supplementary Fig. 2), we presented the discrete strategy in the main text. Although SFC relies on Gaussian assumptions when exploring connectivity between neighbors, it also extracts non-linear information from the relationships between nodes that are not directly connected and separated by multiple relay stations. To obtain a single local and distributed SFC map per subject, we computed the mean of all SFC weighted degree maps that contain the connectivity convergences extracted from each time window. Finally, performing the mean of the 30 subjects, we obtained the local and distributed SFC maps for each of the performed tasks. In these final maps, a larger dynamic SFC degree in a particular voxel indicates higher connectivity streams from the rest of the brain, thus reaching that specific voxel many times, while a smaller degree means low connectivity dynamic convergences.

**Dynamic SFC map association with cortical gene expression**. We used the average SFC maps of all task conditions and the Allen Human Brain Atlas to investigate whether genetic transcription profiles underlie the local and distributed recurrent functional connectivity or attractor-like capabilities of the human brain. The Allen Human Brain Atlas provides whole-brain genome-wide expression patterns for six human subjects[37]. We used a previously generated surface anatomical transformation of the transcriptional profiles of protein-coding genes (20736 genes) based on 58692 measurements of median gene expression in 3702 brain samples[38]. This anatomical transformation is based on the 68 cortical regions of the Desikan–Killiany atlas and covers the entire cortex[53]. First, we converted the average SFC maps of task local and distributed connectivity from the voxel level to 68 Desikan–Killiany regions. We averaged the SFC values of the voxels belonging to each of the 68 cortical regions of the Desikan–Killiany atlas to obtain two vectors describing local and distributed connectivity during task. We used the transcriptional profiles of protein-coding genes to quantify the similarity with our connectivity maps. Second, we investigated the spatial cortical similarity between these SFC maps and cortical expression profiles using GO term analysis with a focus on "Neuro" annotations[54,55] (see Supplementary Tables 3 to 6 for profile details with and without the a priori selection of neuro-related genes). The subset neuro-related genes were obtained from the official tool of the GO Consortium for searching and browsing the GO annotations, AmiGO. The 20736 genes were reduced to 3719 neuro-related genes. The Pearson's correlation approach between the local and distributed vectors and the final list of gene expression vectors was used to evaluate the spatial similarity between then. Third, we built histogram distributions of spatial similarity values to obtain the genetic expression patterns of genes that are significantly associated with the SFC maps. Fourth, we applied an initial statistically significant cutoff (>1.65 SD; based on 90% confidence interval) to obtain a broad list of genes in order to perform GO overrepresentation tests and elucidate the significant functional annotations related to local and distributed attractor-like maps. We used PANTHER13.1 software and Fisher's exact with FDR multiple test correction to perform the statistical testing (FDR at <0.005). We used the GO Biological Process annotation dataset, as we were interested in the investigation of neuro-related biological processes and not in Cellular Component or Molecular Function annotations. Fifth, we used an a priori strategy to investigate the synaptic LTP and LTD genes associated with the regional and global attractor-like maps. We obtained all genes classified as having LTP and LTD functionality from the GO annotation system (AmiGo; LTP = GO:0050806/GO:0060291, positive regulation of synaptic transmission, long-term synaptic potentiation; and LTD = GO:0050805/GO:0060292, negative regulation of synaptic transmission and long-term synaptic depression). Then, we compared the GO LTP and LTD lists with a restricted list of genes related to local and distributed connectivity maps using a Venn diagram and a stringent statistically significant cutoff (>1.96 SD; based on 95% confidence interval) to concisely detect their functional assignment. Finally, we used a regression statistical approach to compare the regression slopes of the spatial associations between the local and distributed connectivity maps and specific candidate genes of LTP and LTD that were functionally detected in the previous step. We used an FWE Bonferroni correction at $p < 0.05$ to correct for multiple comparisons in these contrast analyses.

**Visualization**. Cortical data maps were projected on the brain surface using the population-average landmark- and surface-based (PALS) surface (PALS-B12) provided with Caret software[56]. We used the "interpolated voxel algorithm" and "multi-fiducial mapping" settings and a scale based on weighted degree of SFC to display cortical maps in Figs. 2 and 3. We used Gephi software and an energy layout for the graph visualization in Fig. 3. The energy layout algorithm is a force layout method based on a network energy procedure that arranges the nodes in a network by minimizing the length and crossing of links and optimizing the optimal lengths and positions of them. This network correspond to the combination of the average local and distributed SFC matrices from task-related maps of Fig. 3a, b. We first $z$-score normalized both matrices and to avoid high density of links and improve visualization we plotted all connections above $z > 3.9$ ($p < 0.001$).

**Code availability**. All codes for imaging analysis are available for the research community from the corresponding author (J.S.) upon reasonable request.

## Data availability

All neuroimaging and genetic data that support the findings of this study are available from the Human Connectome Project (https://www.humanconnectomeproject.org) and the Allen Human Brain Atlas (https://human.brain-map.org).

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

## Acknowledgements

This work has been partially supported by the National Institutes of Health (NIH) grant K23EB019023 (to J.S.), Postdoctoral Fellowship Program from the Basque Country Government and Bizkaia Talent (to I.D.).

## Author contributions

Design of the study: J.S.; analysis: I.D. and J.S.; interpretation of the data: I.D. and J.S.; preparation, review or approval of the manuscript: I.D. and J.S.; decision to submit the manuscript for publication: I.D. and J.S.

## Additional information

**Competing interests:** The authors declare no competing interests.

