## [Peer Review File · Nature Communications]

Reviewers' comments:

Reviewer #1 (Remarks to the Author):

Overall

Functional connectivity is now part and parcel of many human neuroimaging studies in cognitive and clinical neuroscience, making it important to improve our understanding of its links with biology. In the past 4-5 years there has also been a push to increase temporal resolutions of acquisition and analysis, by looking at dynamic connectivity rather than static connectivity. This paper aims at finding molecular correlates of local and distributed dynamic functional connectivity in tasks and at rest. This is a problem worth examining, and the use of open data to pursue the question is a strength of the paper. Linking the results here with broader models of cognition is another strong point. While the thrust of the paper is novel, the work is undermined by vagueness in technical details which make it difficult to follow, or reproduce, the results.

I expand on these comments below.

Major points

1) Dynamic connectivity processing. The pipeline described on page 7 does not seem to mention filtering. With a TR of 720 ms, the sampling frequency is 1.39 Hz, and the window chosen contains 21.6 seconds. While I agree in principle with the authors that there is always a tradeoff in choosing window length, for resting-state data at least (and almost certainly for task data), this is too short with respect to the underlying hemodynamic signal, and will lead to spurious fluctuations (See Leonardi et al NeuroImage 2015); this is further exacerbated by the apparent lack of high pass filtering. In addition, the authors estimate in the order of $(5138^2)/2$ correlation coefficients per window, meaning that this is a very high dimensional estimation problem.

Also, note that removing correlations with p-values less than 0.05 is fine in the sense that all thresholds are somewhat arbitrary, but that this gives an erroneous idea of error control - there are of course around $(5138^2)/2$ such hypothesis tests per window, meaning that the actual false positive rate will be much higher than 5%. Indeed, most correlation coefficients here will be false discoveries. See for example figure 2 in Hero and Rajaratnam, J. of the American Statistical Association 2011.

Because this underlies the whole analysis, I would recommend to perform the same analysis again with increased window size, and high pass filtering. Furthermore, I would recommend exploring regularized covariance estimators, in particular see Ledoit and Wolf, J. Multivariate analysis, 2004. Currently it is difficult to trust the rest of the analysis because of this - very high dimensions mean that random noise probably dominates.

2) Dynamic versus 'static' connectivity.

First, On page 10 lines 12-16, the authors mention that performing their 'diffusion' computation within a time window or across time windows yields analogous outcomes. Does this mean that we don't really need the time dimension? Or just that diffusion steps cause rapid decay numerically because of the multiplication?

Second, page 10 line 20, the authors compute the mean of SFC weighted degree maps, thereby removing the time index from their analysis.

Third, In page 14 lines 14-16 the authors state that the local dynamic connectivity map is equivalent to the total connectivity map.

Taken together, these points suggest that what is captured by the 'local attractor-like' dynamics is very similar to static connectivity. It would be good to perform the end-to-end analysis without the windowing (that is, on total connectivity) to explicitly highlight the differences between static and

dynamic connectivity (if any). This would strengthen the authors' case and motivate the use of dynamical systems language like 'attractor' and 'phase space' in the manuscript.

3) Spatial associations mapping

Page 20 lines 10-14 is simply too high-level to understand what was done. This needs many more details, which can go in supplementary if needed.

l.10: How were average SFC maps 'converted' to Desikan-Killiany regional SFC ? By averaging ?

l-11: How was the spatial cortical similarity between SFC maps and Allen gene expression data computed? By averaging gene expression values that fall into each Desikan-Killiany region and computing a Pearson correlation between the vector of such regional expression and the vector of regional SFC ?

If so, are the regional expression and regional SFC vectors Gaussian?

As the manuscript stands it is unfortunately very difficult to judge if the procedure used is valid.

4) GO Enrichment analysis.

One of the main results of the paper is that local connectivity has roughly equal number of over-represented LTP and LTD genes, while distributed connectivity has more LTP genes overexpressed. I congratulate the authors on providing both a results for >1.65 SD and >1.96 SD thresholds on the spatial similarity value (although note that these represent a 90% and 95% confidence interval for Gaussian distributions, which figure 4.I is clearly not). However, enrichment analysis results on GO vary considerably because of its graph structure, meaning that degrees of freedom and independence relationships can be approximately computed only with several assumptions. Because this LTP/LTD balance between the two types of connectivities is a central result of the paper, the authors should provide more details here and perform additional tests.

In particular:

- Please explain how the subset of 3700 "neuro-related genes" was obtained
- Please provide explanation of how enrichment was calculated (test: Fisher hypergeometric? Exact? software: home-made? existing?)
- How many GO:BP categories were tested?
- How does the story change with different significance thresholds on enrichment (e.g. classical Bonferroni 0.01, 0.05 in addition to the more modern but unusual choice of 0.05 presented here) ?
- In addition, please provide expected and observed counts as well as (uncorrected) p-values for each category reported.
- Finally, to provide some insights into the specificity of results (we can expect neuro-related results if we use a neuro-related set of genes), please also run the same analysis with all 16K Allen genes, not only the 3.7K neuro-related genes. Do the same ontology categories rise to the top of enrichment analysis?

5) p.18 lines 24ff: 'Namely, while on one hand... in the local map [...]' is a confusing sentence, please break it up or rephrase.

Minor points

1) p.12, when introducing LTP and LTD GO annotations, please mention which ontology (Biological Process) as well as the specific GO ID you are using (presumably GO:0060291 and GO:0060292).

2) In figure 5.I, the total of genes mentioned in the 'local attractor-like' set (yellow) comes to 211, but the text mentions 193 genes. Is this a mistake or am I misunderstanding the figure ? Likewise for the 'distributed attractor-like' set which comes to 195 while the text mentions 177. I'm assuming there is no overlap and the 18 genes shared between local and distributed are neither

LTP nor LTD genes?

3) In figure 5.II, please provide different line styles or colors for the regressions on local and distributed - it is far from obvious which line belongs to which point cloud

Typos

p. 8 line 20: "nsteps_s" -> "nstep_s"

p.18 lines 19-20: "LDT" -> "LTD" (twice)

Reviewer #2 (Remarks to the Author):

In this paper, Diez & Sepulcre interrogate the dynamic properties of resting state and task fMRI using a combination of graph measures and techniques motivated by dynamic systems theory. They hence report divergent patterns of local versus distributed spatiotemporal "attractors". Intriguingly, these maps covary significantly with maps of synaptic plasticity gene transcription levels. This latter finding, in particular, is quite novel and of likely broad interest to the community.

In its current state, aspects of the study are somewhat uneven and I think it would need a convincing revision, likely involving the analysis of more data, to secure the approval of the community.

Major

1. Methodological: Why only N=30 subjects out of all those available on the HCP (hundreds). The authors should draw more subjects from the HCP for their pipeline, and also use a training and test data set in order to show the generalizability of their analyses.
2. With the use of rs-fMRI, there are invariably methodological issues that need to be addressed: Did the pre-processing pipeline use global signal regression? What was the treatment of head motion and physiological artefacts. Did the authors check for any effects and/or associations with frame-wise head motion?
3. I was surprised to see the analysis proceed at the very high dimensional level of the individual voxel! Was any smoothing applied to the data that could increase the inter-voxel correlations. Why remove negative correlations? How was the thresholding by p-value achieved? Was it done at every time point? What was the sparsity of the ensuing matrices and was this consistent across subjects?
4. Did you regress out the task effects prior to performing the main analysis and use the residuals? Or did you just model the raw (pre-processed) voxel time series?
5. Given the recent controversy in dynamic functional connectivity, the authors should repeat their main analysis on surrogate data generated from the original data using an appropriate multivariate Fourier resampling scheme or auto-regressive model: It is possible much of the structure documented reflects the complex, but stationary spatiotemporal correlations within resting state fMRI data (this concern may be less of a consequence with the clearly dynamic task fMRI).
6. I do really like the analysis method, visualization and interpretation in terms of attractors and convergent zones., However, I don't see the direct relationship to attractors or even low dimensional orbits. What is the meaning of the diverging lines in the "Cumulative Weighted Degree" plots? (Figure 2). How/why do the findings favour a multistability scenario?
7. Is there anything in this analyses that can speak to individual differences (since each analysis is done at the individual subject level) – for example correlations between summary dynamic measures and task performance?
8. Did I miss a step in the analysis description – were the correlations with gene maps performed on the task- or resting state attractors (or their overlap?).

Minor

1. Abstract: Not all neurons are oscillatory (most are stochastic integrators) and certainly not all are synchronized.
2. P3: "... there are no commonly accepted notion about ..." – please make sure to check for consistency of singular versus plural (I won't comment on other instances in the ms)
3. P3: I would avoid use of the term "the human functional connectome" (why only one, and which one?) for "human whole brain functional connectivity"
4. P4 "We wonder whether" suggest reword to "We conjectured that ..."
5. P8-9: I found the pseudo-code provided a suboptimal way of communicating the Methods and suggest that the authors provide a simple mathematical description with relevant definitions notation.
6. P12: How was the Bonferroni correction performed – over genes? Possible associations? Is 0.005 the corrected p-value?
7. P21: There is no value in repeating the material that motivated your study at the beginning of the Discussion: You could delete nearly the entire page 21 and start the Discussion with a succinct summary of your main findings. If you want to re-contextualize your findings, you could do this in the next paragraph (but a short paragraph would suffice).

Reviewer #3 (Remarks to the Author):

The authors applied two network measures, which they call local dynamic connectivity and global (distributed) dynamic connectivity, to open-resource fMRI data. These measures were proposed in previous studies. They particularly found that these measures were specifically correlated with genetic transcription levels of long-term potentiation/depression -related genes.

I do not recommend the publication of this article for two main reasons. (1) I strongly doubt the validity of the network measures they used (despite that they are supported by two published papers). (2) The paper is coarsely written, full of misconceptions and inconsistencies. Please refer to my comments below on these.

[major]

Line 98: "functional streams" What does it mean? I am getting lost already.

Line 98: "degree of recurrent connectivity". The network is recurrent (and undirected, I guess) anyways. So I don't understand what this means.

Line 164: "variance stabilization". This does not make sense. No notion of stability discussed or introduced. What do you mean by stabilizing the variance?

p.8-9: This is the most major point. I doubt the validity of these network measures.

First, LC and DC do not sum to TC. Therefore, LC and DC are not decompositions of the TC (though I don't understand the justification of TC anyways).

Second, on line 182, $step_s$ is defined in terms of $nstep_{s-1}$. It does not define $nstep_s$. Should the LFS be replaced by $nstep_s$?

Third, LC is not a local measure. It takes contribution of paths up to length 6. Even if the effect of triangle is excluded on line 196, I would say this is a global measure, as in 6 steps from voxel i , probably it is possible to reach almost anywhere in the network of 5138 voxels. And why 6 steps?

And in the definition of TC, 7 steps is used. In the same vein, I disagree with the statement "singularity of the inclusion of only local connectivity..." (lines 199-201) as global effects are also there as I stated above. The authors state why they selected 7 steps. But the diameter and mean path length of course depend on networks (particularly the number of ROIs) and particular data.

Line 316: "converged". Where is the notion of convergence? Did the author run some dynamics or an algorithm to assess whether it converges to a certain point or not? Connectivity does not converge. Connectivity is simply a measurement.

Line 321: "dynamic connectivity". Why use this? Static connectivity is not enough? Justification?

Line 323: "global streams of connectivity consistently reached". LC is also global as the authors used 6-7 steps of walks from a seed voxel i .

Line 390: "Recurrent dynamics". This discussion is confusing. This research did not investigate recurrent dynamics. It is about a (dynamic) functional network. If the authors say this is a work on recurrent dynamics, then any functional network studies (say, based on fMRI + network neuroscience) will be studies of recurrent dynamics, which is clearly not the case.

[minor]

Lines 21-22: "oscillatory synchronized neurons". Neurons themselves are not oscillatory (usually). This is a basic.

Lines 22-24: "... neural activity has discovered recurrent dynamic of cerebral microcircuits, it is still poorly understood whether this dynamic principle supports large-scale brain networks." I don't make sense of it. What do you mean by "supporting large-scale brain networks". Networks are simply there (e.g. anatomical networks). Even if one means large-scale functional networks, I don't get what the authors mean by "dynamic principle supports (or not) large-scale brain networks."

Lines 91-92: "see the Methods section for details". This is the methods section.

Line 92: "high modularity". This is underspecified. In network terminology, modularity is community structure. Adjacent areas are not necessarily engage in the same community. If modularity means something else, it is vague as what modularity means is not explained.

Line 108: "N=30". Are you using all participants and the sample size is still this small? If not, selection criteria?

Lines 109-113: The statement is obviously too brief as the results generally depend on how one does preprocessing.

Line 156-157: "a brain mask containing 5138 voxels". Is this a gray matter mask?

Line 171: "network diffusion connectivity". This does not make sense. Why "diffusion"? What's the difference of this to connectivity or edge between a pair of voxels?

Line 172: "dynamics" -> "dynamic"

Lines 185, 199 and 210: Should not be indented.

Line 256: "graph theory changes". Graph theory does not change. Graph theory is a theory.

Line 575: Typo in the first author's name.

Response to Reviewers for *Nature Communications* “**Neurogenetic Profiles Delineate Large-scale Connectivity Dynamics of the Human Brain**” by Diez and Sepulcre.

Reviewer #1 (Remarks to the Author):

Functional connectivity is now part and parcel of many human neuroimaging studies in cognitive and clinical neuroscience, making it important to improve our understanding of its links with biology. In the past 4-5 years there has also been a push to increase temporal resolutions of acquisition and analysis, by looking at dynamic connectivity rather than static connectivity. This paper aims at finding molecular correlates of local and distributed dynamic functional connectivity in tasks and at rest. This is a problem worth examining, and the use of open data to pursue the question is a strength of the paper. Linking the results here with broader models of cognition is another strong point. While the thrust of the paper is novel, the work is undermined by vagueness in technical details, which make it difficult to follow, or reproduce, the results. I expand on these comments below.

Reviewer’s Comment #1 (major):

*Dynamic connectivity processing. The pipeline described on page 7 does not seem to mention filtering. With a TR of 720 ms, the sampling frequency is 1.39 Hz, and the window chosen contains 21.6 seconds. While I agree in principle with the authors that there is always a tradeoff in choosing window length, for resting-state data at least (and almost certainly for task data), this is too short with respect to the underlying hemodynamic signal, and will lead to spurious fluctuations (see Leonardi et al *NeuroImage* 2015); this is further exacerbated by the apparent lack of high pass filtering. In addition, the authors estimate in the order of $(5138^2)/2$ correlation coefficients per window, meaning that this is a very high dimensional estimation problem.*

*Also, note that removing correlations with p-values less than 0.05 is fine in the sense that all thresholds are somewhat arbitrary, but that this gives an erroneous idea of error control - there are of course around $(5138^2)/2$ such hypothesis tests per window, meaning that the actual false positive rate will be much higher than 5%. Indeed, most correlation coefficients here will be false discoveries. See for example figure 2 in Hero and Rajaratnam, *J. of the American Statistical Association* 2011.*

*Because this underlies the whole analysis, I would recommend to perform the same analysis again with increased window size, and high pass filtering. Furthermore, I would recommend exploring regularized covariance estimators, in particular see Ledoit and Wolf, *J. Multivariate analysis*, 2004. Currently it is difficult to trust the rest of the analysis because of this - very high dimensions mean that random noise probably dominates.*

Authors’ Response:

We are extremely thankful to the reviewer for pointing out this important methodological issue of our work. First, we apologize for the missing information about the high-pass filtering (0.01) that was indeed applied in this work. We have amended this point in the

submitted version of our manuscript. Moreover, we agree that the customization of high pass filtering and the window size in dynamic functional connectivity may be more appropriate for this type of work. Thus, following the helpful reviewer’s suggestion, we have used different window lengths and customized high-pass filtering in the new version of the manuscript. Based on *Leonardi et al. NeuroImage, 2015*, window lengths between 30 and 60 seconds seems to be reasonable choice for dynamic functional connectivity, so we performed all new analyses with windows of 30, 45, 50 and 60 seconds. In **Supplementary Table 1**, we show the window size, number of time points in each window and corresponding high pass filter cut-off frequency applied in the new set of analyses. The applied high pass filter cut-off frequency was computed based on the formula:

$$f_{min} = \frac{1}{w}$$

Window Size (seconds)	Number of time points in window	High pass filter cut-off frequency
30	42	0.033
45	63	0.022
50	70	0.02
60	84	0.016

We have also included a new supplementary figure that includes all alternative window sizes (**Supplementary Figure 4**). All corresponding sections have been updated, particularly the methods section as follows:

“In this study, we examined different window lengths and customized high-pass filtering to investigate dynamic connectivity patterns. At the conceptual level, shorter window lengths might provide higher temporal resolution of transient changes but lack the precision to estimate correlation coefficients. Longer window lengths, on the contrary, might improve precision, but the result will tend toward the time-averaged solution. **Supplementary Table 1** show all window sizes used in the analyses, from 30 to 60²⁰. In the main sections of the manuscript, we present findings in which a window size of 30 seconds each (TR=0.72; 42 time points) was used to split the fcMRI data, with 1 lagged time point between them¹⁸. Our overlapping criterion was designed to obtain smooth transitions between network states. Before splitting the time series into different windows of 30 seconds, a high and low pass filter with a cut-off frequency of 0.01 Hz and 0.08 Hz was applied to remove spurious fluctuations^{21,22} (alternative window sizes and high pass filters in **Supplementary Fig. 4**.”

Supplementary Figure 4. Customized high-pass cut-off filtering based on specific window sizes: 30, 45, 50 and 60 seconds.

Moreover, we have followed the reviewer's suggestion about using regularized covariance estimators to obtain connectivity matrices based on *Ledoit and Wolf, J. Multivariate Analysis, 2004*. We have compared our original Pearson-correlation-based matrices with the new regularized-estimation-based matrices. We found that the average correlation between corresponding matrices was of 0.8715 (both matrices were very similar but with lower values for the regularized estimation matrix). Next, we used a density approach to compare the final connectivity maps obtained in **Figure 2**.

We observed that thresholds from 5% to 10% of all possible links in different window sizes result in similar maps (see 10% in figure Below).

Finally, we agree with the reviewer that replication of our findings are important to check if random noise or other type of bias may be influencing our results. Following that suggestion, we have included two new independent datasets (N=30 each) from the Human Connectome Project for replication purposes in the new version of the manuscript. As now seen in **Supplementary Figure 6**, this replication approach shows that our findings are highly reproducible. We have added this information in the new version of the manuscript as follows:

“Apart from the main sample, we included two additional independent samples of 30 individuals each from the Human Connectome Project [both with 17 females and 13 males between 22 and 36 years old] for replication purposes.”

“Similar results were obtained for *local* and *distributed* dynamic connectivity with the two replication datasets (**Supplementary Fig. 6**).”

Supplementary Figure 6. Local and distributed dynamic connectivity patterns of replication datasets 1 and 2. Cortical maps show the average of all task domains analogous to **Figure 3**.

Reviewer's Comment #2 (major):

Dynamic versus 'static' connectivity. First, on page 10 lines 12-16, the authors mention that performing their 'diffusion' computation within a time window or across time windows yields analogous outcomes. Does this mean that we don't really need the time dimension? Or just that diffusion steps cause rapid decay numerically because of the multiplication? Second, page 10 line 20, the authors compute the mean of SFC weighted degree maps, thereby removing the time index from their analysis. Third, in page 14 lines 14-16 the authors state that the local dynamic connectivity map is equivalent to the total connectivity map.

Taken together, these points suggest that what is captured by the 'local attractor-like' dynamics is very similar to static connectivity. It would be good to perform the end-to-end analysis without the windowing (that is, on total connectivity) to explicitly highlight the differences between static and dynamic connectivity (if any). This would strengthen the authors' case and motivate the use of dynamical systems language like 'attractor' and 'phase space' in the manuscript.

Authors' Response:

We appreciate the reviewer's comment and opportunity to clarify this important point in our work. Following this suggestion, we have computed local and distributed connectivity with and without the sliding window approach. As seen in **Supplementary Figure 5**, and in agreement with the reviewer's intuition, the local connectivity maps in the static and dynamic condition is highly similar (average of all subjects and task, as in **Figure 3**). However, we found the distributed connectivity map in the static condition less defined than the distributed connectivity map in the dynamic condition, which advocates for the complementary and specific information of dynamic changes to capture transient distributed connectivity. We have added this new information in **Supplementary Figure 5**.

Supplementary Figure 5. Average maps of local and distributed connectivity patterns in "static" conditions (no sliding window approach).

More importantly, in this study, we aimed to investigate transient connectivity patterns underlying local and distributed brain network states. As recently demonstrated, using simultaneous calcium and hemodynamic fMRI signals (Matsui et al.), it has been shown that short time windows in fMRI represent true transient neuronal co-activations, which we believe give complementary information to the conventional "static" connectivity approach. Although we agree with the reviewer that dynamic and static

functional connectivity may overlap in some features across time scales, we believe the investigation of transient co-activations or connectivity patterns enhances our ability to detect fundamental mechanism of brain functionality. As a mere example of the richness of our dataset in this regard, we have illustrated the brain state transitions of one individual of our sample in the next figure. We computed the mean activation maps and detect the clustering structure of signal fluctuations in 10 different states. We can appreciate how the brain activations maps change with time, yielding different connectivity states.

Moreover, we would like to clarify that we always used the time dimension in our analyses. In graph theory, path analysis is used to investigate how a given node (or vertex) connects to another node by using a sequence of connections (or edges). Path analyses on graphs are commonly referred as graph “diffusion” approaches, due to its ability to capture the spreading of communications across the network from each individual node. Stepwise functional connectivity (SFC) analysis is a graph “diffusion” approach (please note that this is not referring to diffusion MRI). Moreover, SFC can be used to investigate dynamic diffusion patterns of graphs if transient connectivity across time is included in the analysis. Thanks to the reviewers comment and to avoid confusion within the neuroimaging community, we have replaced the term “diffusion” on graphs by path connectivity changes in time or connectivity propagation. Moreover, we would like to remark that there are several ways in which propagation on graphs can be achieved using SFC. In our specific case, we investigated graph-based dynamic changes by using SFC in all time points of our connectivity data in two forms: 1) by calculating the voxel-level path connectivity propagation from the matrix of in each time-point, to later average these results (e.g. $N_1 \times N_1$, $N_1 \times N_1 \times N_1 \dots$ then $N_2 \times N_2$, $N_2 \times N_2 \times N_2 \dots$), or 2) by calculating the voxel-level path connectivity propagation from consecutive matrices (e.g. $N_1 \times N_2$, $N_1 \times N_2 \times N_3 \dots$ then $N_2 \times N_3$, $N_2 \times N_3 \times N_4 \dots$). Both strategies yielded extremely similar results (**Supplementary Figure 2**).

In a related note, we used the mean of SFC weighted degree maps in all our figures to show the average time-domain behavior. By adopting this final representation of our findings, we do not attempt to eliminate the dynamical information of our findings but to highlight the central tendency of connectivity transitions in the spatial cortical maps. In order to complement the average time-domain representations, we show the time-index information in the line graphs of **Figure 2**.

Finally, the reviewer is correct regarding the local and total dynamic connectivity maps. We have found that the spatial distribution of local dynamic connectivity changes overlaps those when all connectivity is under consideration (without segregating local and distributed from the total connectivity). We would like to highlight that our SFC results using local and total connectivity are not equal in numbers but they show similar spatial distributions (please see also **Reviewer's Comment #4** of **Reviewer #3** in this regard). The spatial similarity arises from the fact that local and total connectivity data display similar modular structure on their respective graphs. On contrast, the distributed connectivity data (in which closed neighborhoods have been mathematically removed) shows a distinctive pattern that arises from a non-modular structure on its graph. As it is well known, task states increase local modularity in fcMRI data, therefore, it is not surprising that our local condition shows similar spatial distribution than the total condition, while the distributed condition shows a unique pattern of non-modular connectivity.

Reviewer's Comment #3 (major):

Spatial associations mapping. Page 12 lines 10-14 is simply too high-level to understand what was done. This needs many more details, which can go in supplementary if needed. l.10: How were average SFC maps 'converted' to Desikan-Killiany regional SFC? By averaging? l-11: How was the spatial cortical similarity between SFC maps and Allen gene expression data computed? By averaging gene expression values that fall into each Desikan-Killiany region and computing a Pearson correlation between the vector of such regional expression and the vector of regional SFC? If so, are the regional expression and regional SFC vectors Gaussian?

As the manuscript stands it is unfortunately very difficult to judge if the procedure used is valid.

Authors' Response:

Thank to the reviewer's comment, we have added a more detailed explanation of the methodology in the text (see "*Spatial Associations Between Dynamic SFC Maps and Cortical Gene Expression*" section). The SFC maps were converted into the Desikan-Killiany atlas by averaging the SFC values of the 68 regions of interest. Then, we used a Pearson correlation between the vector of regional SFC and the vectors of regional gene expression to compute the spatial similarity scores. As the SFC and gene expression values have or approach a Gaussian distribution (see below), we opted for a Pearson correlation approach.

Reviewer's Comment #4 (major):

GO Enrichment analysis. One of the main results of the paper is that local connectivity has roughly equal number of over-represented LTP and LTD genes, while distributed connectivity has more LTP genes overexpressed. I congratulate the authors on providing both a results for >1.65 SD and >1.96 SD thresholds on the spatial similarity value (although note that these represent a 90% and 95% confidence interval for Gaussian distributions, which figure 4.1 is clearly not). However, enrichment analysis results on GO vary considerably because of its graph structure, meaning that degrees of freedom and independence relationships can be approximately computed only with several assumptions. Because this LTP/LTD balance between the two types of connectivities is a central result of the paper, the authors should provide more details here and perform additional tests.

In particular:

- *Please explain how the subset of 3700 "neuro-related genes" was obtained.*
- *Please provide explanation of how enrichment was calculated (test: Fisher hypergeometric? Exact? software: home-made? existing?)*
- *How many GO: BP categories were tested?*
- *How does the story change with different significance thresholds on enrichment (e.g. classical Bonferroni 0.01, 0.05 in addition to the more modern but unusual choice of 0.05 presented here)?*
- *In addition, please provide expected and observed counts as well as (uncorrected) p-values for each category reported.*
- *Finally, to provide some insights into the specificity of results (we can expect neuro-related results if we use a neuro-related set of genes), please also run the same analysis with all 16K Allen genes, not only the 3.7K neuro-related genes. Do the same ontology categories rise to the top of enrichment analysis?*

Authors' Response:

We appreciate the reviewer noticing this missing information in the previous version of the manuscript. The subset of 3719 "neuro-related genes" was obtained from the official tool of the Gene Ontology Consortium for searching and browsing the GO annotations, AmiGO. We used AmiGO to select all genes previously characterized as a neuro-related role by GO annotations. Moreover, we used PANTHER13.1 software and the Fisher's exact with FDR multiple test correction to perform the statistical testing. Note that we used the GO Biological Process annotation dataset in this analysis. Other GO annotation datasets such as Cellular Component or Molecular Function were not involved in the study, as we were only interested in the investigation of neuro-related biological processes. Following the reviewer's suggestion, we have provided the full profile of our GO analyses, including uncorrected and corrected p-values and fold enrichment in **Supplementary Table 2** and **Supplementary Table 3**. As we used a very strict statistical threshold correction, we observed that all relevant biological processes highlighted in our results remain the same if other p-values cut-off are selected. Finally, we have also provided a new supplementary material (**Supplementary Table 4** and **Supplementary Table 5**) that shows GO analysis with the entire profile of genes (without the a priori selection of neuro-related genes). Using this strategy, we found a degree of overlap with our previous findings and also the presence of new biological processes associated to basic DNA/RNA processes. We

have added a detailed explanation of each specific point in the corresponding sections of the new manuscript.

“We used the average SFC maps of all task conditions and the Allen Human Brain Atlas to investigate whether genetic transcription profiles underlie the *local* and *distributed* recurrent functional connectivity or attractor-like capabilities of the human brain. The Allen Human Brain Atlas provides whole-brain genome-wide expression patterns for six human subjects¹⁶. We used a previously generated surface anatomical transformation of the transcriptional profiles of protein-coding genes (20,736 genes) based on 58,692 measurements of median gene expression in 3,702 brain samples¹⁷. This anatomical transformation is based on the 68 cortical regions of the Desikan-Killiany atlas and covers the entire cortex²⁸. First, we converted the average SFC maps of task *local* and *distributed* connectivity from the voxel-level to 68 Desikan-Killiany regions. We averaged the SFC values of the voxels belonging to each of each 68 cortical regions of the Desikan-Killiany atlas to obtain 2 vectors describing *local* and *distributed* connectivity during task. We used the transcriptional profiles of protein-coding genes to quantify the similarity with our connectivity maps. Second, we investigated the spatial cortical similarity between these SFC maps and cortical expression profiles using Gene Ontology (GO) term analysis with a focus on “Neuro” annotations^{29,30} (see **Supplementary Tables 2 to 5** for profile details with and without the a priori selection of neuro-related genes). The subset neuro-related genes were obtained from the official tool of the GO Consortium for searching and browsing the GO annotations, AmiGO. The 20,736 genes were reduced to 3,719 neuro related genes. A Pearson correlation approach between the *local* and *distributed* vectors and the final list of gene expression vectors was used to evaluate the spatial similarity between them. Third, we built histogram distributions of spatial similarity values to obtain the genetic expression patterns of genes that are significantly associated with the SFC maps. Fourth, we applied an initial statistically significant cutoff (>1.65 SD) to obtain a broad list of genes in order to perform GO overrepresentation tests and elucidate the significant functional annotations related to *local* and *distributed* attractor-like maps. We used PANTHER13.1 software and the Fisher's exact with FDR multiple test correction to perform the statistical testing (FDR at <0.005). We used the GO *Biological Process* annotation dataset, as we were interested in the investigation of neuro-related biological processes and not in *Cellular Component* or *Molecular Function* annotations. Fifth, we used an a priori strategy to investigate the synaptic long-term potentiation (LTP) and long-term depression (LTD) genes associated with the local and global attractor-like maps. We obtained all genes classified as having LTP and LTD functionality in the GO analysis. Then, we compared the GO LTP and LTD lists with a restricted list of genes related to *local* and *distributed* connectivity maps using a Venn diagram and a stringent statistically significant cutoff (>1.96 SD) to concisely detect their functional assignment. Finally, we used a regression statistical approach to compare the regression slopes of the spatial associations between the *local* and *distributed* connectivity maps and specific candidate genes of LTP and LTD that were functionally detected in the previous step. We used an FWE *Bonferroni* correction at $p < 0.05$ to correct for multiple comparisons in these contrast analyses.”

Reviewer's Comment #5 (major):

p.18 lines 24ff: 'Namely, while on one hand... in the local map [...]' is a confusing sentence, please break it up or rephrase.

Authors' Response:

Following the reviewer's recommendation we have rephrase that sentence as follows:

"Namely, we found that the spatial relationships between the *local* map and KCNB1 (LTD) and SYT12 (LTP) genes are significantly different than the same spatial relationships with the *distributed* map ($p=0.0011$ and $p=0.0002$, respectively). Moreover, the spatial relationship between the *distributed* map and the PRNP gene (LTP) is significantly different than the same spatial relationship with the *local* map ($p=0.0031$)."

Reviewer's Comment #6 (minor):

p.12, when introducing LTP and LTD GO annotations, please mention which ontology (Biological Process) as well as the specific GO ID you are using (presumably GO:0060291 and GO:0060292).

Authors' Response:

We agree with the reviewer that specific GO IDs are needed in this section to define our GO analysis of *Biological Process* search. We added this information in the methods section as follows:

"We used the GO *Biological Process* annotation dataset, as we were interested in the investigation of neuro-related biological processes and not in *Cellular Component* or *Molecular Function* annotations."

"Fifth, we used an a priori strategy to investigate the synaptic long-term potentiation (LTP) and long-term depression (LTD) genes associated with the regional and global attractor-like maps. We obtained all genes classified as having LTP and LTD functionality from the GO annotation system (AmiGo; LTP=GO:0050806/GO:0060291, positive regulation of synaptic transmission, long-term synaptic potentiation; and LTD=GO:0050805/GO:0060292, negative regulation of synaptic transmission and long-term synaptic depression)."

Reviewer's Comment #7 (minor):

In figure 5.1, the total of genes mentioned in the 'local attractor-like' set (yellow) comes to 211, but the text mentions 193 genes. Is this a mistake or am I misunderstanding the figure? Likewise for the 'distributed attractor-like' set which comes to 195 while the text mentions 177. I'm assuming there is no overlap and the 18 genes shared between local and distributed are neither LTP nor LTD genes?

Authors' Response:

We thank the reviewer for identifying this unnoticed mistake. In the previous version of the text, we mentioned the non-overlapping genes between the "local attractor-like" the "distributed attractor-like". This information has been updated to include the total number of genes in each profile. Finally, the reviewer is also correct with her/his second comment. The 18-shared genes between the "local attractor-like" the "distributed attractor-like" are neither LTP nor LTD genes.

Reviewer's Comment #8 (minor):

In figure 5.II, please provide different line styles or colors for the regressions on local and distributed - it is far from obvious which line belongs to which point cloud.

Authors' Response:

Thanks to the reviewer's comment, we have changed the colors of the line of the linear fit in both scatterplots.

Reviewer's Comment #9 (minor):

p. 8 line 20: "nsteps_s" -> "nstep_s"

Authors' Response:

We thank the reviewer for detecting this typo that has been corrected in the new version of the manuscript.

Reviewer's Comment #10 (minor):

p.18 lines 19-20: "LDT" -> "LTD" (twice)

Authors' Response:

We thank the reviewer for detecting this typo that has been corrected in the new version of the manuscript.

Reviewer #2 (Remarks to the Author):

In this paper, Diez & Sepulcre interrogate the dynamic properties of resting state and task fMRI using a combination of graph measures and techniques motivated by dynamic systems theory. They hence report divergent patterns of local versus distributed spatiotemporal “attractors”. Intriguingly, these maps co-vary significantly with maps of synaptic plasticity gene transcription levels. This latter finding, in particular, is quite novel and of likely broad interest to the community.

In its current state, aspects of the study are somewhat uneven and I think it would need a convincing revision, likely involving the analysis of more data, to secure the approval of the community.

Reviewer’s Comment #1 (major):

Methodological: Why only N=30 subjects out of all those available on the HCP (hundreds). The authors should draw more subjects from the HCP for their pipeline, and also use a training and test data set in order to show the generalizability of their analyses.

Authors’ Response:

Although at first our study was restricted to one dataset of 30 individuals due to the high computational demands of the dynamic analysis, we have followed the reviewer’s suggestion and have included two new independent datasets (N=30 each) from the human connectome project for replication and generalization purposes in the new version of the manuscript. It is important to notice that our sliding window method generates 2.812 connectivity matrices per subject (84.360 matrices in total) and the high dimensionality of these matrices increase drastically the computation time for local and distributed connectivity. Even if high performance systems were used to run the analysis in parallel to aid processing time, it takes a couple of weeks to run all the analysis. As now seen in **Supplementary Figure 6**, our replication approach with two alternative datasets shows a high degree of reproducibility of our findings. We have added this information in the new version of the manuscript as follows:

“Apart from the main sample, we included two additional independent samples of 30 individuals each from the Human Connectome Project [both with 17 females and 13 males between 22 and 36 years old] for replication purposes.”

“Similar results were obtained for *local* and *distributed* dynamic connectivity with the two replication datasets (**Supplementary Fig. 6**).”

Supplementary Figure 6. Local and distributed dynamic connectivity patterns of replication datasets 1 and 2. Cortical maps show the average of all task domains analogous to **Figure 3**.

Reviewer's Comment #2 (major):

With the use of rs-fMRI, there are invariably methodological issues that need to be addressed: Did the pre-processing pipeline use global signal regression? What was the treatment of head motion and physiological artifacts. Did the authors check for any effects and/or associations with frame-wise head motion?

Authors' Response:

We appreciate the reviewer's comment and opportunity to clarify this important point in our work. In the new version of the manuscript we have updated the **Supplementary Material** with the requested information about the pre-processing pipeline used in this study. In short, we used FSL and AFNI to pre-process the fMRI data. First, the fMRI dataset was aligned to the middle volume, using a six-parameter (rigid body) linear transformation, to correct for head movement artifacts; the transformation matrix of each volume to the middle volume was used to compute 24 motion parameters. After intensity normalization, the 24 motion parameters, the average cerebrospinal fluid (CSF) signal and the average white-matter signal were regressed out, followed by the removal of linear and quadratic trends. Thus, global signal regression was not used in this study. Next, the functional data was spatially normalized to the MNI152 brain template, with a voxel size of 3*3*3 mm³ and smoothed with a 6 mm full width at half maximum (FWHM) isotropic Gaussian kernel. Finally, a down sampling to 8 mm was applied to compute graph analysis method at the voxel level. We did not check for frame-wise displacement effects as we expected to remove the effect of motion artifacts when doing the average across the windows and subjects.

Reviewer's Comment #3 (major):

I was surprised to see the analysis proceed at the very high dimensional level of the individual voxel! Was any smoothing applied to the data that could increase the inter-voxel correlations? Why remove negative correlations? How was the thresholding by p-

value achieved? Was it done at every time point? What was the sparsity of the ensuing matrices and was this consistent across subjects?

Authors' Response:

Thanks to the reviewer's comment (also brought by **Reviewer #1**), we have investigated these important methodological steps and free parameters associated to them to ensure that our findings are reproducible and stable. As mentioned in the previous point, the functional data was spatially normalized to the MNI152 brain template, smoothed with a 6 mm full width at half maximum (FWHM) isotropic Gaussian kernel, and down sampling to 8 mm in order to be able to compute our highly demanding graph analysis method at the voxel level (please keep in mind that our approach generates thousands of connectivity matrices per subject). We are expecting that all these steps increase the inter-voxel correlations, to a certain degree, as voxels keep retaining individual information. Moreover, all the connections with negative correlation values or correlation values with a p-value less than 0.05 were removed from the functional matrices to eliminate the network links with poor interpretability and low temporal correlation in the context of graph theory. For instance, as previously reported, negative correlations can emerge in a brain graph as a mere epiphenomenon of the network distance topology between node pairs (Chen et al., 2011). Finally, we used a sliding window approach in which each Pearson correlation-based connectivity matrix was thresholded at p-value < 0.05 to eliminate low temporal correlations. This threshold leads to an average of 8.27% of all possible links in each window. As the percentage of links used in each time window slightly differs for different windows and subjects, we adopted two alternative strategies. First, to ensure the results are the same with a consistent sparsity in the matrices across all windows and subjects, we performed the same analysis but instead of removing links with p-value < 0.05 we took different levels of link density (e. g. at the 10% of highest connectivity links, see figure below). We observed that thresholds from 5% to 10% of all possible links in different window sizes resulted in similar maps.

Second, we followed the suggestion of reviewer 1 about using regularized covariance estimators to obtain connectivity matrices based on *Ledoit and Wolf, J. Multivariate Analysis, 2004*. We compared our original Pearson-correlation-based with the new regularized-estimation-based matrices and found that the average correlation between corresponding matrices was of 0.8715 (both matrices were very similar but with lower values for the regularized estimation matrix).

Reviewer's Comment #4 (major):

Did you regress out the task effects prior to performing the main analysis and use the residuals? Or did you just model the raw (pre-processed) voxel time series?

Authors' Response:

We appreciate the reviewer's comment about this important point of the preprocessing pipeline. In this study, we use the raw (pre-processed) data for all analyses as we were in search of common dynamic connectivity patterns associated to all tasks in natural scenarios, without manipulating the input signal. This is explained as follows:

“Conventional fcMRI approaches derive connectivity information from the entirety of the BOLD time series and result in a time-averaged brain network graph. However, as brain network dynamic changes occur at a higher temporal scale, other strategies have been used to take full advantage of the non-stationarities that reside in temporal information contained in the fcMRI data¹⁸. For instance, the sliding window approach extracts the dynamic interactions between brain areas by using a time moving-window along the BOLD time series. As demonstrated by a recent study using simultaneous calcium and hemodynamic signals, short time windows represent transient neuronal co-activation that allows the capture of more information about different brain states compared to static connectivity¹⁹. In this study, we examined different window lengths and customized high-pass filtering to investigate dynamic connectivity patterns. At the conceptual level, shorter window lengths might provide higher temporal resolution of transient changes but lack the precision to estimate correlation coefficients. Longer window lengths, on the contrary, might improve precision, but the result will tend toward the time-averaged solution. **Supplementary Table 1** show all window sizes used in the analyses, from 30 to 60²⁰. In the main sections of the manuscript, we present findings in which a window size of 30 seconds each (TR=0.72; 42 time points) was used to split the fcMRI data, with 1 lagged time point between them¹⁸...Before splitting the time series into different windows of 30 seconds, a high and low pass filter with a cut-off frequency of 0.01 Hz and 0.08 Hz was applied to remove spurious fluctuations^{21,22} (alternative window sizes and high pass filters in **Supplementary Fig. 4**; static condition, no sliding window approach **Supplementary Fig. 5**). The Pearson correlation of the time series of all the voxels in each time window was computed, which generated a functional connectivity matrix for each time window. We used a whole brain mask -containing gray matter, subcortical structures and cerebellum- of 5,138 voxels to extract the BOLD time series and applied the sliding window approach...”

Reviewer's Comment #5 (major):

Given the recent controversy in dynamic functional connectivity, the authors should repeat their main analysis on surrogate data generated from the original data using an appropriate multivariate Fourier resampling scheme or auto-regressive model: It is possible much of the structure documented reflects the complex, but stationary spatiotemporal correlations within resting state fMRI data (this concern may be less of a consequence with the clearly dynamic task fMRI).

Authors' Response:

We really thank the reviewer for this excellent suggestion. Following the reviewers comment we have repeated the main analyses of this work using surrogate data

generated from method 3 in Hurtado et al 2004 (*Statistical method for detection of phase locking episodes in neural oscillations*). This method scrambles the phase spectrum of signals whilst preserving the amplitude spectrum. We obtained the distribution of the original and surrogate data for local (blue) and distributed (red). The panels below represent the distribution using mean local and mean distributed maps for all task and subjects. These distributions show that original and surrogate data display distinctive patterns.

Most importantly, surrogate data present high similarity between the spatial maps of distributed and local connectivity (correlation of 0.93, left scatterplot below) while real data show completely different maps (correlation of -0.21, right scatterplot below).

Reviewer’s Comment #6 (major):

I do really like the analysis method, visualization and interpretation in terms of attractors and convergent zones. However, I don’t see the direct relationship to attractors or even low dimensional orbits. What is the meaning of the diverging lines in the “Cumulative Weighted Degree” plots? (Figure 2). How/why do the findings favor a multistability scenario?

Authors’ Response:

The reviewer is correct. We agree with the reviewer that more support was needed to fully support our interpretation about the “attractorness”. In the previous version of the

manuscript, we focused on the dynamic connectivity patterns that display recurrent and convergent behavior in the brain network, which are incidental evidence of attractor-like behavior. In the current version of our work, we have performed additional analyses that deepen and confirm the attractor nature of our findings. As our SFC approach can describe the dynamic trajectories of connectivity on graphs (SFC is a graph theory propagation metric), we were able to detect where these trajectories target. We took advantage of the description of convergent zones of the original sample and described the trajectories of dynamic connectivity in a replication sample. If our interpretation of the dynamic connectivity follows an attractor behavior, we expect to see two phenomena: 1) trajectories converging toward specific networks and 2) trajectories remaining inside those networks (rather than leaving those networks toward other parts of the brain). Indeed, we found that dynamic paths not only converge toward specific networks of the human brain but also once in those networks the dynamic connectivity remains consistently within them. In other words, once in a convergent zone the trajectories remain there, and if not in a convergent zone the trajectory goes toward them. This is a more direct measure of attractorness. Therefore, thanks to the reviewer we have included a new figure showing this result.

“Finally, as our SFC approach is able describe dynamic trajectories of connectivity on graphs, we also analyzed the trajectories of dynamic connectivity using the original and replication datasets. We evaluated the cortical areas with specific SFC values and test if dynamic trajectories of paths remain inside or go outside those areas. We found that cortical areas with high *local* and *distributed* dynamic connectivity (or SFC values) tend to display dynamic paths that remain repeatedly inside those areas, while regions with low SFC values display dynamic paths that go toward cortical areas with high SFC values (**Fig. 4**)”

Figure 4. Description of dynamic trajectories of paths from cortical areas discovered as *local* and *distributed* connectivity cores during task performance. Masks from cortical areas with predominant (**Figure 3-III**; blue/magenta in cortical masks) and non-predominant (**Figure 3-III**; none blue/magenta colors in cortical masks) *local* (left) and *distributed* (right) connectivity were used to obtain the amount of dynamic trajectories that remain inside or leave outside these connectivity cores in an independent sample of individuals. “*LC to LC*” and “*DC to DC*” refers to connectivity trajectories that start and end within the predominant/core areas of local and distributed connectivity. “*LC to Rest*” and “*DC to Rest*” refers to connectivity trajectories that start in the predominant/core areas but end outside them. “*Rest to LC*” and “*Rest to DC*” refers to connectivity trajectories that start outside the predominant/core areas but end inside them. Error bars represent standard error. Y-axes represent the weighted degree of *local* or *distributed* paths (normalized by the size of cortical masks).

Regarding the other comments of the reviewer, line plots in Figure 2 show the accumulation of SFC values in each voxel, meaning the accumulation of paths or streams that reach that specific voxel repeatedly. Now this clarification is added in the results section.

“... (cortical maps in Fig. 2-I; line graphs in Fig. 2-I; note that line graphs represent the SFC values or connectivity paths that reach specific voxels repeatedly).”

Finally, we believe our findings on *local* and *distributed* attractor networks favors a multi-stability scenario because we found that multiple and not a unique attractor network dominates the connectivity landscape of the human brain dynamics. Although the existence of one network for the *distributed* connectivity may support that the DMN is the main stabilizer of connections at the large-scale level.

“Our findings favor a multi-stability scenario of the human whole brain functional connectivity in which the temporal patterns of connectivity tend to converge into specific points of the connectome space at the *local* and *distributed* level. Importantly, they also support the notion that the DMN plays an important role as a global attractor-like network across cognitive states, in which network configurations in heteromodal cortices confer dynamic properties in search of stability when large distances and distributed connectivity are engaged. Therefore if only distributed connectivity is under consideration, the human brain may tend to display a uni- rather than a multi-stable dynamic system.”

Reviewer’s Comment #7 (major):

Is there anything in this analyses that can speak to individual differences (since each analysis is done at the individual subject level) - for example correlations between summary dynamic measures and task performance?

Authors’ Response:

We agree with the reviewer that the study of individual differences is a really important topic for the field. Our study was designed to capture dynamic connectivity patterns that underlay different task performances. In other words, we aimed to search for common patterns of transient functional connectivity across multiple tasks and independently of subjects. We agree that our analytical approach will be also able to distinguish individual features if the appropriate transformation of the design is done. For instance, we feel this transformation will require additional data, such as highly sampled individuals, which was not included in this data. Therefore, we believe the investigation of individual differences in the present study would fall out of scope but has a high potential for future studies.

Reviewer's Comment #8 (major):

Did I miss a step in the analysis description - were the correlations with gene maps performed on the task- or resting state attractors (or their overlap?).

Authors' Response:

We thank the reviewer for the opportunity to clarify this point in the text. The correlations with gene maps were performed with the mean connectivity maps of all the subjects and tasks (not on the resting maps as we were interested in the common dynamic connectivity patterns across task conditions). We have highlighted this information in the methods section of the manuscript as follows:

“First, we converted the average SFC maps of task *local* and *distributed* connectivity from the voxel-level to 68 Desikan-Killiany regions. We averaged the SFC values of the voxels belonging to each of each 68 cortical regions of the Desikan-Killiany atlas to obtain two vectors describing *local* and *distributed* connectivity during task. We used the transcriptional profiles of protein-coding genes to quantify the similarity with our connectivity maps.”

Reviewer's Comment #9 (minor):

Abstract: Not all neurons are oscillatory (most are stochastic integrators) and certainly not all are synchronized.

Authors' Response:

This point has been amended in the abstract.

Reviewer's Comment #10 (minor):

P3: "...there are no commonly accepted notion about..." please make sure to check for consistency of singular versus plural (I won't comment on other instances in the ms)

Authors' Response:

Thanks to the reviewer's comment we have checked for consistency of singular/plural and language use across the manuscript.

Reviewer's Comment #11 (minor):

P3: I would avoid use of the term “the human functional connectome” (why only one, and which one?) for “human whole brain functional connectivity”

Authors’ Response:

Following the reviewer’s comments we have modified the term “the human functional connectome” as suggested.

Reviewer’s Comment #12 (minor):

P4 “We wonder whether” suggest reword to “We conjectured that …;”

Authors’ Response:

Following the reviewer’s suggestion we have modified this sentence.

Reviewer’s Comment #13 (minor):

P8-9: I found the pseudo-code provided a suboptimal way of communicating the Methods and suggest that the authors provide a simple mathematical description with relevant definitions notation.

Authors’ Response:

Thanks to the reviewer’s comment we have provided a detailed mathematical description of our approach in the main text of the manuscript as follows:

Total connectivity (TC) for node i is computed as:

$$S_1(i, j) = \frac{c(i, j) - \min(c)}{\max(c) - \min(c)}$$

$$S_s(i, j) = \sum_{k=1}^n \frac{S_{s-1}(i, k) - \min(S_{s-1})}{\max(S_{s-1}) - \min(S_{s-1})} \frac{c(k, j) - \min(c)}{\max(c) - \min(c)} \quad [i \neq j, s > 1]$$

$$TC(i) = \sum_{s=1}^7 \sum_{j=1}^n \frac{S_s(i, j) - \min(S_s)}{\max(S_s) - \min(S_s)}$$

where c is the association connectivity matrix, n is the number of nodes (voxels) in association connectivity matrix and S_s represents the normalized stepwise connectivity matrix for step s .

Local connectivity (LC) for node i is computed as:

$$SL_1(i, j) = \frac{c(i, j) - \min(c)}{\max(c) - \min(c)} [S_2(i, j) \neq 0]$$

$$SL_s(i, j) = \sum_{k=1}^n \frac{SL_{s-1}(i, k) - \min(SL_{s-1})}{\max(SL_{s-1}) - \min(SL_{s-1})} \frac{c(k, j) - \min(c)}{\max(c) - \min(c)} \quad [i \neq j, s > 1]$$

$$LC(i) = \sum_{s=1}^7 \sum_{j=1}^n \frac{SL_s(i, j) - \min(SL_s)}{\max(SL_s) - \min(SL_s)}$$

Where c is the association connectivity matrix, n is the number of nodes (voxels) in association connectivity matrix and SL_s represents the normalized local stepwise connectivity matrix for step s .

Distributed connectivity (DC) for node i is computed as:

$$SD_1(i, j) = S_4(i, j) \quad [S_1(i, j) = S_2(i, j) = 0]$$

$$SD_s(i, j) = \sum_{k=1}^n \frac{SD_{s-1}(i, k) - \min(SD_{s-1})}{\max(SD_{s-1}) - \min(SD_{s-1})} \frac{c(k, j) - \min(c)}{\max(c) - \min(c)} \quad [i \neq j, s > 1]$$

$$DC(i) = \sum_{s=1}^6 \sum_{j=1}^n \frac{SD_s(i, j) - \min(SD_s)}{\max(SD_s) - \min(SD_s)}$$

Where c is the association connectivity matrix, n is the number of nodes (voxels) in association connectivity matrix and SD_s represents the normalized distributed stepwise connectivity matrix for step s .

Reviewer's Comment #14 (minor):

P12: How was the Bonferroni correction performed - over genes? Possible associations? Is 0.005 the corrected p-value?

Authors' Response:

We apologize for this omission. We used FWE correction in analyses of Figure 5. The section involving Gene Ontology overrepresentation analysis was corrected using FDR instead of FWE. Now this is amended in the text. Moreover, we have provided more detailed information about the GO overrepresentation analysis. For instance, we used PANTHER13.1 software and the Fisher's Exact with FDR multiple test correction to perform the statistical testing (q level at <0.005). Therefore, no arbitrary/subjective criteria were introduced in this process.

Reviewer's Comment #15 (minor):

P21: There is no value in repeating the material that motivated your study at the beginning of the Discussion: You could delete nearly the entire page 21 and start the Discussion with a succinct summary of your main findings. If you want to re-contextualize your findings, you could do this in the next paragraph (but a short paragraph would suffice).

Authors' Response:

Following the reviewer's suggestion, we have modified the entire first section of the discussion as follows:

"In this study, we studied empirical functional connectivity data over time during various cognitive states to reveal the zones of the human brain in which convergence of recurrent connectivity occurs. Our aim was to detect and map the specific areas engaged in attractor-like behavior at the *local* and *distributed* level. As previously suggested, we assumed that the human brain has numerous recurrent dynamic sources and networks that may produce multiple attractors in a multi-stability scenario^{8,40}. Neural activity does not occur in isolation but is synchronized with other neuronal signals. This organization tends to repeat over time, and some coupled regions tend to be orchestrated more frequently and recurrently than others. Connectivity between brain areas via phase synchronization forms functional networks, and dynamic and transient patterns arise from the cooperation and competitiveness among them. Moreover, recurrent activity across neural networks is thought to yield self-organized and multi-scaled dynamics patterns in the human brain^{2,32-34}. At the spatial level, it has been demonstrated that flows of activity spread from specific areas toward certain local or distant locations of the cortex. In general, this property of brain activity streaming repeatedly toward precise locations can be seen or conceptualized as an attractor or attractor-like behavior. In the past, neural network modeling has reproduced feasible scenarios of recurrent or attractor-like dynamic patterns at the synaptic and neuronal levels. Since the introduction of the concept put forth by Lorente de No and Hebb of reverberation as neural activity that reiterates in a network, researchers have studied the implications of recurrent neuronal activity in cell assembly formation and cellular memory processes in brain circuits, which are key components of cortical networks³⁵. However, it is still poorly understood how these types of dynamics transfer or generalize to larger spatial scales and whether self-organized patterns, such as reverberancy/recurrence or attractor-like behavior, arise from the functional connections of the human cortex. The existence of self-organized and attractor dynamic patterns in the human brain networks has been postulated to be critical to our understanding of how cognitive processes, behavior, action-perception cycles or mind-brain-body integration forms in humans^{32,36,37}. Compared to previous studies, our study employed a data-driven approach that goes directly from empirical fcMRI data to the investigation of the biological basis supporting large-scale dynamic patterns of the human brain. By doing so, we show that the DMN displays dynamic connectivity and genetic features that favors it as the main attractor network of the human brain."

Reviewer #3 (Remarks to the Author):

The authors applied two network measures, which they call local dynamic connectivity and global (distributed) dynamic connectivity, to open-resource fMRI data. These measures were proposed in previous studies. They particularly found that these measures were specifically correlated with genetic transcription levels of long-term potentiation/depression-related genes.

I do not recommend the publication of this article for two main reasons. (1) I strongly doubt the validity of the network measures they used (despite that they are supported by two published papers). (2) The paper is coarsely written, full of misconceptions and inconsistencies. Please refer to my comments below on these.

Reviewer's Comment #1 (major):

Line 98: "functional streams" What does it mean? I am getting lost already.

Authors' Response:

We thank the reviewer for the opportunity to clarify this important point of our work. "Stream" is a widely used and well-established term in the field of system neuroscience (Milner and Goodale, 1992). It refers to the pathways by which the brain communicates different large-scale systems. As such we have adapted this term from previous literature to accommodate the functional connectivity and graph theory frameworks. To avoid confusion in some readers, thanks to the reviewer's comment, we have replaced the "functional stream" term by more conventional terms such as path analysis on graphs.

Reviewer's Comment #2 (major):

Line 98: "degree of recurrent connectivity". The network is recurrent (and undirected, I guess) anyways. So I don't understand what this means.

Authors' Response:

In order to avoid the confusion pointed out by the reviewer, we have replaced this sentence by "degree of paths that repeatedly reach each node of the brain" (please see **Reviewer's Comment #7**).

Reviewer's Comment #3 (major):

Line 164: "variance stabilization". This does not make sense. No notion of stability discussed or introduced. What do you mean by stabilizing the variance?

Authors' Response:

We apologize for the misleading information of these two terms. In the preprocessing pipeline, we used a Fisher transformation of our data. This is a common step in functional connectivity neuroimaging. The Fisher transformation is a variance-stabilizing transformation for correlation coefficients. As a consequence, the variance of a Fisher corrected correlation matrix is approximately constant for all values of the population correlation coefficient ρ . Without the Fisher transformation, the variance of

the correlation coefficients grows smaller as $|\rho|$ gets closer to 1. To avoid confusion with these terms, we have rephrased this sentence as follows:

“Finally, we applied a variance-stabilizing transformation (Fisher transformation) to all correlation coefficients of association connectivity matrices as a final step before our graph theory based analysis (c in equation Condition 1 to 3)²⁰.”

Reviewer’s Comment #4 (major):

p.8-9: This is the most major point. I doubt the validity of these network measures. First, LC and DC do not sum to TC. Therefore, LC and DC are not decompositions of the TC (though I don't understand the justification of TC anyways).

Authors’ Response:

The LC network and the DC network sum the TC network. LC network includes the triangle motifs of the TC network, and the DC network includes the non-triangle components of the TC network (thus, LC network + DC network= TC network). The SFC analysis shows that when triangle motifs are included in the analysis (LC and TC networks), the spatial distributions of SFC values are similar. If the triangle motifs are removed (DC network), the spatial distribution of SFC values is different. In other words, the triangle motifs dominate the network structure not only in LC (as an obvious results of our mathematical constrain) but also TC. We believe it is important to show the TC condition in order to show the original network structure in which LC and DC is based upon.

Reviewer’s Comment #5 (major):

Second, on line 182, $step_s$ is defined in terms of $nstep_{\{s-1\}}$. It does not define $nstep_s$. Should the LFS be replaced by $nstep_s$?

Authors’ Response:

Thank for the reviewer for pointing out to this unnoticed mistake. In the current version of the manuscript we have fixed this issue (please see formula in the methods section).

Reviewer’s Comment #6 (major):

Third, LC is not a local measure. It takes contribution of paths up to length 6. Even if the effect of triangle is excluded on line 196, I would say this is a global measure, as in 6 steps from voxel i , probably it is possible to reach almost anywhere in the network of 5138 voxels. And why 6 steps? And in the definition of TC, 7 steps is used. In the same vein, I disagree with the statement "singularity of the inclusion of only local connectivity..." (lines 199-201) as global effects are also there as I stated above. The authors state why they selected 7 steps. But the diameter and mean path length of course depend on networks (particularly the number of ROIs) and particular data.

Authors’ Response:

It is important to remark that our local and distributed terms relates to network-based topology and not to Euclidean distances within the human brain (now this is clarified in the main text). Our LC approach is based on a network topology that only takes into account triangle motifs on the graphs. The density of triangle motifs in a graph is a direct property of a local organization of the graph (a typical motif defining segregated communities). The number of steps used to explore this type of graphs does not affect

their local condition. In other words, regardless of the length of paths, the network structure is always local, and the result of a path analysis with a propagation/diffusion method will always advance through a local neighborhood of nodes (or triangle motifs). For instance, if one runs a random walk to infinite number of steps in a local structure of triangle motifs, it will remain local all the time (unless a non-local/non-triangle structure is reached). So, it is not a matter of the step lengths but the structure. **Supplementary Figure 1** shows an example of *local* and *distributed* connectivity from a V1 seed. As seen in this figure, our measure of *local* connectivity from V1 shows a diffusion pattern that remains in the local neighborhood. Moreover, as the reviewer acknowledged, the use of 7 steps is well justified in the paper, based on our previous work with similar brain mask, network density and mean path length (Sepulcre et al. Journal of Neuroscience, 2012). Of note, the *distributed* connectivity condition only used 6 steps (not 7 as LC) because we have removed the motifs involved in the first step, making impossible to do count on it (please see formula for details).

Reviewer's Comment #7 (major):

Line 316: "converged". Where is the notion of convergence? Did the author run some dynamics or an algorithm to assess whether it converges to a certain point or not? Connectivity does not converge. Connectivity is simply a measurement.

Authors' Response:

Stepwise functional connectivity (SFC) analysis is a graph theory approach for path analysis on graphs. When SFC is applied to dynamic connectivity data, it detects the dynamic paths that reach every single node in the network. Thus, if a region displays a high degree of SFC across time, it means that many nodes in the brain network converge in it. This is now better represented in **Figure 1, 2 and 4**, and mathematically expressed in the new formulas of the main text.

Reviewer's Comment #8 (major):

Line 321: "dynamic connectivity". Why use this? Static connectivity is not enough? Justification?

Authors' Response:

We appreciate the reviewer's comment and opportunity to clarify this important point in our work (thanks also to **Reviewer #1**). In the new version of the manuscript, we have computed local and distributed connectivity with and without the sliding window approach. As seen in **Supplementary Figure 5**, and in agreement with the reviewer's intuition, the local connectivity maps in the static and dynamic condition is similar (average of all subjects and task, as in **Figure 3**). However, we found the distributed connectivity map in the static condition less defined than the distributed connectivity map in the dynamic condition, which advocate for the complementary and specific information of dynamic changes to capture transient distributed connectivity. Moreover, it is important to remark that this information only refers to the final spatial display of a minimal part of our findings, which would be impossible to obtain without the dynamic evaluation (**Figure 2 to 4**).

Supplementary Figure 5. Average maps of local and distributed connectivity patterns in "static" conditions (no sliding window approach).

Reviewer's Comment #9 (major):

Line 323: "global streams of connectivity consistently reached". LC is also global as the authors used 6-7 steps of walks from a seed voxel i.

Authors' Response:

Please see response to **Reviewer's Comment #6**.

Reviewer's Comment #10 (major):

Line 390: "Recurrent dynamics". This discussion is confusing. This research did not investigate recurrent dynamics. It is about a (dynamic) functional network. If the authors say this is a work on recurrent dynamics, then any functional network studies (say, based on fMRI + network neuroscience) will be studies of recurrent dynamics, which is clearly not the case.

Authors' Response:

The reviewer is correct. We agree with the reviewer that more support was needed to fully support our interpretation about the recurrent dynamics. In the previous version of the manuscript, we focused on the path analysis based on dynamic connectivity patterns that display convergent behavior in the brain network, which are incidental evidence of recurrent and attractor-like behavior. In the current version of our work, we have performed additional analyses that deepen and confirm the recurrent nature of our findings. As our SFC approach can describe the dynamic trajectories of connectivity on graphs (SFC is a graph theory propagation metric), we were able to detect where these trajectories target. We took advantage of the description of convergent zones of the original sample and described the trajectories of dynamic connectivity in a replication sample. If our interpretation of the dynamic connectivity follows an attractor behavior, we expect to see two phenomena: 1) trajectories converging toward specific networks and 2) trajectories remaining inside those networks (rather than leaving those networks toward other parts of the brain). Indeed, we found that dynamic paths not only converge toward specific networks of the human brain but also once in those networks the dynamic connectivity remains consistently within them. In other words, once in a convergent zone the trajectories remain there, and if not in a convergent zone the trajectory goes toward them. This is a more direct measure of recurrent and attractor dynamics. Therefore, thanks to the reviewer we have included a new figure showing this result.

“Finally, as our SFC approach is able describe dynamic trajectories of connectivity on graphs, we also analyzed the trajectories of dynamic connectivity using the original and replication datasets. We evaluated the cortical areas with specific SFC values and test if dynamic trajectories of paths remain inside or go outside those areas. We found that cortical areas with high *local* and *distributed* dynamic connectivity (or SFC values) tend to display dynamic paths that remain repeatedly inside those areas, while regions with low SFC values display dynamic paths that go toward cortical areas with high SFC values (Fig. 4)”

Figure 4. Description of dynamic trajectories of paths from cortical areas discovered as *local* and *distributed* connectivity cores during task performance. Masks from cortical areas with predominant (Figure 3-III; blue/magenta in cortical masks) and non-predominant (Figure 3-III; none blue/magenta colors in cortical masks) *local* (left) and *distributed* (right) connectivity were used to obtain the amount of dynamic trajectories that remain inside or leave outside these connectivity cores in an independent sample of individuals. “LC to LC” and “DC to DC” refers to connectivity trajectories that start and end within the predominant/core areas of local and distributed connectivity. “LC to Rest” and “DC to Rest” refers to connectivity trajectories that start in the predominant/core areas but end outside them. “Rest to LC” and “Rest to DC” refers to connectivity trajectories that start outside the predominant/core areas but end inside them. Error bars represent standard error. Y-axes represent the weighted degree of *local* or *distributed* paths (normalized by the size of cortical masks).

Reviewer’s Comment #11 (minor):

Lines 21-22: "oscillatory synchronized neurons". Neurons themselves are not oscillatory (usually). This is a basic.

Authors' Response:

Thanks to the reviewer's comment we have amended the text accordingly.

Reviewer's Comment #12 (minor):

Lines 22-24: "...neural activity has discovered recurrent dynamic of cerebral microcircuits, it is still poorly understood whether this dynamic principle supports large-scale brain networks." I don't make sense of it. What do you mean by "supporting large-scale brain networks". Networks are simply there (e.g. anatomical networks). Even if one means large-scale functional networks, I don't get what the authors mean by "dynamic principle supports (or not) large-scale brain networks."

Authors' Response:

We appreciate the reviewer's comment. Following this suggestion, we have modified this sentence to refine its meaning.

"Experimental and modeling work of neural activity has discovered recurrent and attractor dynamic of cerebral microcircuits. However, it is still poorly understood whether similar dynamic principles exist or can be generalizable to the large-scale level."

Reviewer's Comment #13 (minor):

Lines 91-92: "see the Methods section for details". This is the methods section.

Authors' Response:

Thank to the reviewer for the comment, we have updated the text in this regard.

Reviewer's Comment #14 (minor):

Line 92: "high modularity". This is underspecified. In network terminology, modularity is community structure. Adjacent areas are not necessarily engage in the same community. If modularity means something else, it is vague as what modularity means is not explained.

Authors' Response:

We agree with the reviewer that this term was not explained and may be vague in the context of the present study. We have opted to remove it from the main text to avoid misinterpretations.

Reviewer's Comment #15 (minor):

Line 108: "N=30". Are you using all participants and the sample size is still this small? If not, selection criteria?

Authors' Response:

Although at first our study was restricted to one dataset of 30 individuals -randomly selected from the human connectome project- due to the high computational demands of the dynamic analysis, we have followed the reviewer’s suggestion and have included two new independent datasets (N=30 each) from the human connectome project for replication and generalization purposes in the new version of the manuscript. It is important to notice that our sliding window method generates 2.812 connectivity matrices per subject (84.360 matrices in total) and the high dimensionality of these matrices increase drastically the computation time for local and distributed connectivity. Even if high performance systems were use to run the analysis in parallel to aid processing speed, it takes a couple of weeks to run all the analysis. As now seen in **Supplementary Figure 6**, our replication approach with two alternative datasets shows a highly reproducibility of our findings. We have added this information in the new version of the manuscript as follows:

“Apart from the main sample, we included two additional independent samples of 30 individuals each from the Human Connectome Project [both with 17 females and 13 males between 22 and 36 years old] for replication purposes.”

“Similar results were obtained for *local* and *distributed* dynamic connectivity with the two replication datasets (**Supplementary Fig. 6**).”

Supplementary Figure 6. Local and distributed dynamic connectivity patterns of replication datasets 1 and 2. Cortical maps show the average of all task domains analogous to **Figure 3**.

Reviewer’s Comment #16 (minor):

Lines 109-113: The statement is obviously too brief as the results generally depend on how one does preprocessing.

Authors’ Response:

Thanks to the reviewer's comment, corresponding details of the preprocessing have been added to supplementary material.

Reviewer's Comment #17 (minor):

Line 156-157: "a brain mask containing 5138 voxels". Is this a gray matter mask?

Authors' Response:

We used a whole brain mask that contains gray matter, subcortical structures and cerebellum. We have updated this information in the main text.

Reviewer's Comment #18 (minor):

Line 171: "network diffusion connectivity". This does not make sense. Why "diffusion"? What's the difference of this to connectivity or edge between a pair of voxels?

Authors' Response:

We appreciate the reviewer's comment and opportunity to clarify this important point in our work. In graph theory, path analysis is used to investigate how a given node (or vertex) connects to another node by using a sequence of connections (or edges). Path analyses on graphs can be also referred as "diffusion" on graph analyses, due to its ability to capture the "spreading" or "propagation" patterns of connectivity from each individual node to the rest of the network. Stepwise functional connectivity (SFC) analysis is a graph "diffusion" approach (please note that this is not referring to diffusion MRI). Moreover, SFC can be used to investigate dynamic diffusion patterns of graphs if transient connectivity across time is included in the analysis. Thanks to the reviewers comment and to avoid confusion within the neuroimaging community, we have replaced the term "diffusion" on graphs by path changes in time. Moreover, we would like to remark that there are several ways in which dynamic path analysis on graphs can be achieved using SFC. In our specific case, we investigated graph-based dynamic changes by using SFC in all time points of our connectivity data in two forms: 1) by calculating the voxel-level path connectivity propagation from the matrix of in each time-point, to later average these results (e.g. $N1*N1$, $N1*N1*N1$...then $N2*N2$, $N2*N2*N2$...), or 2) by calculating the voxel-level path connectivity propagation from consecutive matrices (e.g. $N1*N2$, $N1*N2*N3$...then $N2*N3$, $N2*N3*N4$...). Both strategies yielded extremely similar results (**Supplementary Figure 2**).

Reviewer's Comment #19 (minor):

Line 172: "dynamics" -> "dynamic"

Authors' Response:

Thanks to the reviewer's comment this issue has been corrected in the new version of the manuscript.

Reviewer's Comment #20 (minor):

Lines 185, 199 and 210: Should not be indented.

Authors' Response:

Thanks to the reviewer's comment this issue has been corrected in the new version of the manuscript.

Reviewer's Comment #21 (minor):

Line 256: "graph theory changes". Graph theory does not change. Graph theory is a theory.

Authors' Response:

Thanks to the reviewer's comment this issue has been corrected in the new version of the manuscript.

Reviewer's Comment #22 (minor):

Line 575: Typo in the first author's name.

Authors' Response:

Thanks to the reviewer's comment this issue has been corrected in the new version of the manuscript.

REVIEWERS' COMMENTS:

Reviewer #1 (Remarks to the Author):

I thank the authors for their thoughtful and thorough revision. The sensitivity analysis on window size and regularization gives more confidence in the results, the replication on HCP data is nice, and the whole paper is much stronger now.

This is a neat study and I have no further comments.

Reviewer #2 (Remarks to the Author):

The authors have been responsive to the prior concerns and the manuscript is substantially improved. Further edits and justifications are listed below. The authors should also consider their findings in light of a similar paper by Saggari et al. (see point 13, below).

1. Line 22: Please edit: "Experimental and modeling work of neural activity has discovered recurrent and attractor dynamic of cerebral microcircuits. "
2. Line 50: Delete "the" from "supported by the human whole brain functional connectivity"
3. Line 53: Simplify this sentence, "Models based on experimental work have revealed computational properties of neurons such as their recurrent –neural circuits forming directed cycles and exhibiting repetitive temporal dynamics- and attractor –a dynamic pattern that a system tends to evolve or settle into- behaviors"
4. Line 59, "Findings from electroencephalogram (EEG) neurophysiological experiments have also pointed to the existence of recurrent, reverberant or attractor patterns—such as limit cycle and fixed-point attractors—that in turn might explain the large-scale multi-stable synchronicity of the brain." – suggest to delete "neurophysiological" and replace the citation [6] with Freyer, et al. (2011). Journal of Neuroscience, 31, 6353-6361. As this paper deals more explicitly with the content of the sentence.
5. Line 64: Citation 6 (original or new one) is not relevant to seizures; suggest to add Jirsa, et al (2014). Brain, 137(8), 2210-2230.
6. Line 157: Suggest to change "shorter window lengths might provide higher temporal resolution of transient changes but lack the precision to estimate correlation coefficients. Longer window lengths, on the contrary, might improve precision, but the result will tend toward the time-averaged solution." to "shorter window lengths provide higher temporal resolution but the estimated correlation coefficients are noisy and prone to error. Longer window lengths, on the contrary, might yield more precise estimates, but lack temporal fidelity and tend toward the time-averaged solution." and cite the following relevant papers,

Leonardi et al. (2015) NeuroImage, 104:430-436
Zalesky et al. (2015) Neuroimage, 114, 466-470.
7. Line 171: I assume this is a "whole brain gray matter mask"
8. Paragraph beginning line 180: The authors should bear in mind the difference between structural connectivity (hubs, rich clubs, brain networks) and the type of functional connectivity which is employed in the present study.
9. Line 324: What is the rationale for the choice of >1.65 SD for the "initial statistically significant

cutoff"? Why not use a formal sparsity regularizer? Line 338, under what criteria os >1.96 SD "stringent"?

10. Line 387: Similar results were obtained ... with the two replication data sets – could there be a simple way of quantifying this, such as the Dice coefficient?

11. Figure 3.III. What are the x- and y-axes of this figure?

12. Line 459 – again remove "the"

13. p23: I suggest the authors consider convergence of their findings with the recently published paper of a similar vein, Saggari et al. (2018) Nature communications, 9(1), 1399.

14. Line 498: Should be "structurally stable"

REVIEWERS' COMMENTS:

Reviewer #1 (Remarks to the Author):

I thank the authors for their thoughtful and thorough revision. The sensitivity analysis on window size and regularization gives more confidence in the results, the replication on HCP data is nice, and the whole paper is much stronger now.

This is a neat study and I have no further comments.

Reviewer #2 (Remarks to the Author):

The authors have been responsive to the prior concerns and the manuscript is substantially improved. Further edits and justifications are listed below. The authors should also consider their findings in light of a similar paper by Saggat et al. (see point 13, below).

1. Line 22: Please edit: "Experimental and modeling work of neural activity has discovered recurrent and attractor dynamic of cerebral microcircuits. "
2. Line 50: Delete "the" from "supported by the human whole brain functional connectivity"
3. Line 53: Simplify this sentence, "Models based on experimental work have revealed computational properties of neurons such as their recurrent –neural circuits forming directed cycles and exhibiting repetitive temporal dynamics- and attractor –a dynamic pattern that a system tends to evolve or settle into- behaviors"
4. Line 59, "Findings from electroencephalogram (EEG) neurophysiological experiments have also pointed to the existence of recurrent, reverberant or attractor patterns—such as limit cycle and fixed-point attractors⁶—that in turn might explain the large-scale multi-stable synchronicity of the brain." – suggest to delete "neurophysiological" and replace the citation [6] with Freyer, et al. (2011). Journal of Neuroscience, 31, 6353-6361. As this paper deals more explicitly with the content of the sentence.
5. Line 64: Citation 6 (original or new one) is not relevant to seizures; suggest to add Jirsa, et al (2014). Brain, 137(8), 2210-2230.
6. Line 157: Suggest to change "shorter window lengths might provide higher temporal resolution of transient changes but lack the precision to estimate correlation coefficients. Longer window lengths, on the contrary, might improve precision, but the result will tend toward the time-averaged solution." to "shorter window lengths provide higher temporal resolution but the estimated correlation coefficients are noisy and prone to error. Longer window lengths, on the contrary, might yield more precise estimates, but lack temporal fidelity and tend toward the time-averaged solution." and cite the following relevant papers,
Leonardi et al. (2015) NeuroImage, 104:430-436
Zalesky et al. (2015) Neuroimage, 114, 466-470.
7. Line 171: I assume this is a "whole brain gray matter mask"
8. Paragraph beginning line 180: The authors should bear in mind the difference between structural connectivity (hubs, rich clubs, brain networks) and the type of functional connectivity which is employed in the present study.
9. Line 324: What is the rationale for the choice of >1.65 SD for the "initial statistically significant cutoff"? Why not use a formal sparsity regularizer? Line 338, under what criteria os >1.96 SD "stringent"?
10. Line 387: Similar results were obtained ... with the two replication data sets – could there be a simple way of quantifying this, such as the Dice coefficient?
11. Figure 3.III. What are the x- and y-axes of this figure?
12. Line 459 – again remove "the"
13. p23: I suggest the authors consider convergence of their findings with the recently published paper of a similar vein, Saggat et al. (2018) Nature communications, 9(1), 1399.
14. Line 498: Should be "structurally stable"

REVIEWERS' COMMENTS:

Reviewer #1 (Remarks to the Author):

I thank the authors for their thoughtful and thorough revision. The sensitivity analysis on window size and regularization gives more confidence in the results, the replication on HCP data is nice, and the whole paper is much stronger now.

This is a neat study and I have no further comments.

Reviewer #2 (Remarks to the Author):

The authors have been responsive to the prior concerns and the manuscript is substantially improved. Further edits and justifications are listed below. The authors should also consider their findings in light of a similar paper by Saggari et al. (see point 13, below).

Reviewer's Comment #1 Line 22: Please edit: "*Experimental and modeling work of neural activity has discovered recurrent and attractor dynamic of cerebral microcircuits.*"

Authors' Response: Following the reviewer's suggestion, we have modified this sentence as follows:

"Experimental and modeling work of neural activity has described recurrent and attractor dynamic patterns in cerebral microcircuits."

Reviewer's Comment #2 Line 50: Delete "the" from "supported by the human whole brain functional connectivity"

Authors' Response: Thanks to the reviewer's comment, we have fixed this point in the text.

Reviewer's Comment #3 Line 53: Simplify this sentence, "Models based on experimental work have revealed computational properties of neurons such as their recurrent –neural circuits forming directed cycles and exhibiting repetitive temporal dynamics- and attractor –a dynamic pattern that a system tends to evolve or settle into-behaviors"

Authors' Response: Following the reviewer's suggestion, we have simplified this sentence as follows:

"Models based on experimental work have revealed several dynamic properties of neurons. Neural circuits forming directed cycles exhibit repetitive or recurrent temporal dynamics and attractor behaviors –a dynamic pattern that a system tends to evolve or settle into-."

Reviewer's Comment #4 Line 59, "Findings from electroencephalogram (EEG) neurophysiological experiments have also pointed to the existence of recurrent, reverberant or attractor patterns—such as limit cycle and fixed-point attractors⁶—that in turn might explain the large-scale multi-stable synchronicity of the brain." – suggest to

delete “neurophysiological” and replace the citation [6] with Freyer, et al. (2011). Journal of Neuroscience, 31, 6353-6361. As this paper deals more explicitly with the content of the sentence.

Authors’ Response: We thank the reviewer’s for this comment. Freyer et al. citation has been added to the manuscript and the “neurophysiological” term has been removed.

Reviewer’s Comment #5 Line 64: Citation 6 (original or new one) is not relevant to seizures; suggest to add Jirsa, et al (2014). Brain, 137(8), 2210-2230.

Authors’ Response: We appreciate the reviewer’s comment. In the new version of the text we have replaced citation 6 by Jirsa, et al (2014). Brain, 137(8), 2210-2230.

Reviewer’s Comment #6 Line 157: Suggest to change “shorter window lengths might provide higher temporal resolution of transient changes but lack the precision to estimate correlation coefficients. Longer window lengths, on the contrary, might improve precision, but the result will tend toward the time-averaged solution.” to “shorter window lengths provide higher temporal resolution but the estimated correlation coefficients are noisy and prone to error. Longer window lengths, on the contrary, might yield more precise estimates, but lack temporal fidelity and tend toward the time-averaged solution.” and cite the following relevant papers,

Leonardi et al. (2015) NeuroImage, 104:430-436

Zalesky et al. (2015) Neuroimage, 114, 466-470.

Authors’ Response: We are extremely thankful to the reviewer’s suggestion and have updated the manuscript accordantly.

Reviewer’s Comment #7 Line 171: I assume this is a “whole brain gray matter mask”

Authors’ Response: The reviewer is correct. As stated in the main text, we used a gray matter mask containing cortical gray matter, subcortical structures and cerebellum.

Reviewer’s Comment #8 Paragraph beginning line 180: The authors should bear in mind the difference between structural connectivity (hubs, rich clubs, brain networks) and the type of functional connectivity which is employed in the present study.

Authors’ Response:

Following the reviewer’s suggestion, we have modified this sentence as follows:

“...they have increased our knowledge of the structural and functional hierarchies and hubs organization (cortical core of hubs and rich club) that integrate large-scale networks in the human brain^{12,15,30–32}”.

Reviewer’s Comment #9 Line 324: What is the rationale for the choice of >1.65 SD for

the “initial statistically significant cutoff”? Why not use a formal sparsity regularizer?
Line 338, under what criteria is >1.96 SD “stringent”?

Authors’ Response: We thank the reviewer for this insightful comment. In this study, we have developed a novel neuroimaging-genetics approach based on normally distributed coefficients of similarity between spatial maps of genetic and connectivity information. In order to keep our strategy as conventional as possible, we decided to use a standard statistical approach based on confidence intervals of normally distributed variables. The first statistically significant cutoff was set to 90% of the confidence interval ($\mu+1.65SD$) and the second was set to 95% of the confidence interval ($\mu+1.96SD$). We have clarified this point in the main text.

Reviewer’s Comment #10 Line 387: Similar results were obtained ... with the two replication data sets – could there be a simple way of quantifying this, such as the Dice coefficient?

Authors’ Response: We appreciate the reviewer’s comment. In order to maintain consistency with our neuroimaging-genetic strategy in which we obtained similarity values based on linear (Pearson) approach, we decided to measure the spatial similarity between functional connectivity maps using a linear correlation strategy.

We obtained for local connectivity:

Original dataset vs Replication 1: 0.978

Original dataset vs Replication 2: 0.9805

We obtained for distributed connectivity:

Original dataset vs Replication 1: 0.903

Original dataset vs Replication 2: 0.90

We have updated the manuscript with the mean similarity for local and distributed connectivity.

Reviewer’s Comment #11 Figure 3.III. What are the x- and y-axes of this figure?

Authors’ Response: In the submitted version of Figure 3-III, we displayed a network layout based on spring forces. This figure does not include any x- or y-axes, but we will be delighted to fix any related issue if this is a misunderstanding of the reviewer’s comment on our side.

Reviewer’s Comment #12 Line 459 – again remove “the”

Authors’ Response: Thanks to the reviewer’s comment, we have removed this word from line 459.

Reviewer’s Comment #13 p23: I suggest the authors consider convergence of their findings with the recently published paper of a similar vein, Saggari et al. (2018) Nature communications, 9(1), 1399.

Authors' Response: We agree with the reviewer that both works are complementary. However, as our work focused on the cortical distribution of dynamic patterns and Saggari et al. work focused on the graph space level, we found it quite difficult to establish the common ground between both. Following the reviewer's suggestion, we have referenced this important work in the main text.

Reviewer's Comment #14 Line 498: Should be "structurally stable"

Authors' Response: Thanks to the reviewer's comment, we have updated the manuscript accordingly.